# Ultra-high Resolution Watermarking Framework Resistant to Extreme Cropping and Scaling

**Nan Sun**[1], **LuYu Yuan**[1], **Han Fang**[2], **Yuxing Lu**[3], ***Hefei Ling**[1], **Sijing Xie**[1], **Chengxin Zhao**[1]

[1]School of Computer Science and Technology, Huazhong University of Science and Technology
[2]National University of Singapore
[3]Peking University
{sunnan, yuanly, lhefei,xiesijing,chengxinzhao }@hust.edu.cn
fanghan@nus.edu.sg, yxlu0613@gmail.com

## Abstract

Recent developments in DNN-based image watermarking techniques have achieved impressive results in protecting digital content. However, most existing methods are constrained to low-resolution images as they need to encode the entire image, leading to prohibitive memory and computational costs when applied to high-resolution images. Moreover, they lack robustness to distortions prevalent in large-image transmission, such as extreme scaling and random cropping. To address these issues, we propose a novel watermarking method based on implicit neural representations (INRs). Leveraging the properties of INRs, our method employs resolution-independent coordinate sampling mechanism to generate watermarks pixel-wise, achieving ultra-high resolution watermark generation with fixed and limited memory and computational resources. This design ensures strong robustness in watermark extraction, even under extreme cropping and scaling distortions. Additionally, we introduce a hierarchical multi-scale coordinate embedding and a low-rank watermark injection strategy to ensure high-quality watermark generation and robust decoding. Experimental results show that our method significantly outperforms existing schemes in terms of both robustness and computational efficiency while preserving high image quality. Our approach achieves an accuracy greater than 98% in watermark extraction with only 0.4% of the image area in 2K images. These results highlight the effectiveness of our method, making it a promising solution for large-scale and high-resolution image watermarking applications.

## 1 Introduction

With the rapid progress of the digital age, images have become a fundamental medium of information exchange, reaching unprecedented scales in dissemination and application across various fields. Concurrently, image watermarking techniques have emerged as pivotal tools for copyright protection, data security, and integrity verification. However, with the increasing demand for processing large-scale and high-resolution images, DNN-based watermarking approaches face significant challenges in adapting to the requirements of large-scale image watermarking.

Typical deep learning-based image watermarking methods are generally designed for low-resolution images, requiring full-image processing (Zhu et al., 2018; Tancik et al., 2020a; Fang et al., 2022). As a result, these methods face significant limitations when applied to ultra-high resolution (UHR) images. First, processing the entire UHR image incurs high computational costs, resulting in long processing times and potential memory overflow, which impacts efficiency and feasibility. Second, in the decoding phase, UHR images are more vulnerable to scaling and cropping distortions during

---

*Corresponding Author (lhefei@hust.edu.cn)

transmission. Existing watermarking methods, optimized for low-resolution content, struggle to preserve watermark integrity under these severe transformations.

As shown in Figure 1 (a), existing methods for embedding information in high-resolution images typically use a block-based approach (Guo et al., 2023), where the image is divided into non-overlapping blocks, and the same watermark is embedded into each block. However, this approach has two main drawbacks. First, the accuracy of block localization is crucial for decoding performance; misalignment can significantly reduce extraction effectiveness. Second, because the watermark block is smaller than the original image, it is highly susceptible to scaling distortions. An alternative approach embeds the watermark on low-resolution images and then interpolates the residuals to higher resolutions for embedding (Bui et al., 2023). While this reduces the computational burden, it also makes the watermark more vulnerable to local cropping attacks, compromising its robustness.

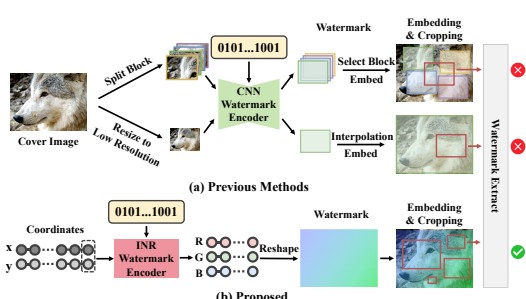

Figure 1: Different high resolution watermark embedding schemes. Where the color blocks on the image represent the watermark embedding area and the red boxes represent the area where the image is cropped for decoding the watermark.

To address the challenges of watermarking large-scale images, we propose an innovative solution based on Implicit Neural Representations (INRs). As illustrated in Figure 1 (b), our method maps continuous pixel coordinates directly to the corresponding RGB values of the watermark. This eliminates the constraint of fixed image resolutions, enabling watermark embedding in UHR images while ensuring robustness against extreme cropping and scaling distortions. The main contributions of this paper can be summarized as follows:

- We propose an innovative INR-based framework for ultra-high resolution watermarking, offering a groundbreaking solution to the challenges of watermarking high-resolution images.
- We introduce a hierarchical multi-Scale coordinates embedding mechanism for accurate watermark generation across scales. In addition, we introduce a low-rank injection scheme for efficient integration of watermarks.
- Extensive experiments on widely representative datasets demonstrate the exceptional performance and significant advantages of our proposed method in handling high-resolution images and accommodating diverse resolution scenarios. In addition, our method exhibits excellent resistance to extreme cropping and scaling that often occurs in high-resolution images.

## 2    Related Work

**DNN-based Image Watermarking.** Recent advances in deep learning have significantly impacted digital image watermarking. HiDDeN (Zhu et al., 2018) introduced an end-to-end DNN-based watermarking framework, resembling an autoencoder, setting the stage for future models. StegaStamp (Tancik et al., 2020a) improves print-shooting robustness by simulating the printing and photographing process. RIHOOP (Jia et al., 2020) further refines this by introducing a differentiable distortion model that preserves the integrity of the watermark under camera imaging conditions. Subsequent works (Jia et al., 2021; Fang et al., 2023; Li et al., 2024; Sun et al., 2024) focus on increasing robustness against various distortions. However, all of these methods are limited to fixed low-resolution images (typically less than $512$). As the resolution of images increases, these methods encounter substantial challenges, including a significant rise in computational complexity and memory usage, which not only slow down processing times but also make their practical application with high-resolution images increasingly difficult.

**High-Resolution Image Watermarking.** Several approaches have been proposed to address the challenges of watermark embedding in high-resolution images. DWSF (Guo et al., 2023) uses a block-based strategy, selecting fixed-size watermark blocks for embedding, but it struggles with block localization and scaling resistance. TrustMark (Bui et al., 2023), on the other hand, generates

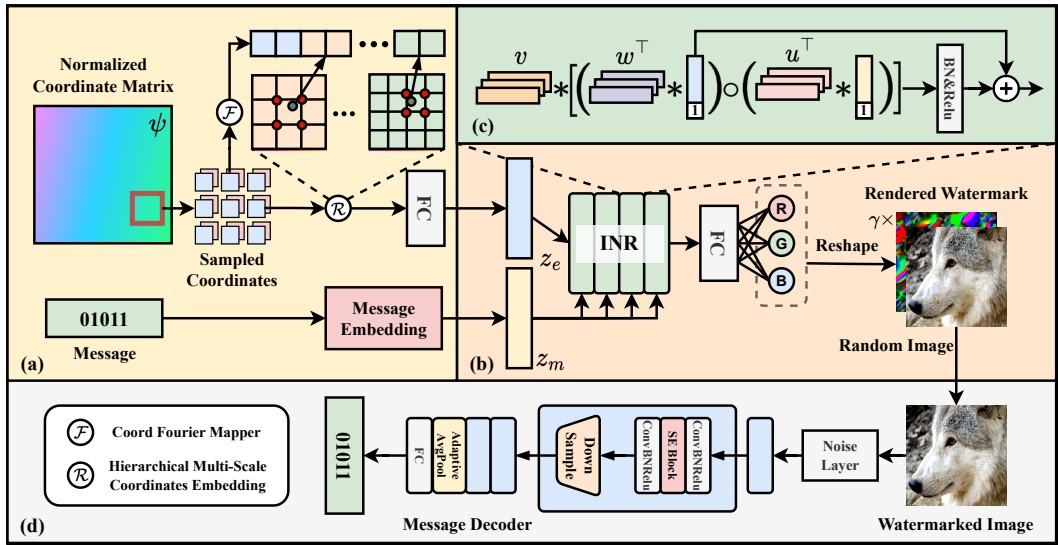

Figure 2: Architecture of the proposed model. (a) shows the process of embedding coordinates and messages, and (b) shows the process of rendering into watermarks via INR. (c) shows the process of low-rank watermark injection. (d) shows the decoding process of our method.

watermark residues on low-resolution images and then uses linear interpolation to scale them to high-resolution images. However, this approach is vulnerable to local cropping attacks. Wang et al. (Wang et al., 2024) use Implicit Neural Representations to fit the host image and then fine-tune the INR to embed watermark information at various resolutions. However, this method has significant drawbacks, as it requires the training of separate networks for each individual image. This process not only increases the computational load but also makes it highly time-consuming, particularly when dealing with large datasets, limiting its practicality for real-time applications. Although the above methods attempt to embed watermarks in large-sized images, none of them effectively balance high-rato cropping, scaling resilience, and real-time performance, which are key challenges in large image watermarking.

**Implicit Neural Representations.** Implicit Neural Representations (INRs) use deep neural networks to model continuous mappings between inputs and outputs, rather than relying on predefined rules. INRs have been widely applied in 3D reconstruction (Mildenhall et al., 2021; Gafni et al., 2021; Hui et al., 2024), super-resolution (Chen et al., 2021; Yang et al., 2021; Chen et al., 2022), and image generation (Skorokhodov et al., 2021; Shaham et al., 2021; Anokhin et al., 2021). In the image domain, an INR takes spatial coordinates as input and outputs RGB values, representing the image as a continuous signal. CNN-based watermarking methods struggle with large images due to high memory and computation costs. In contrast, INRs model continuous signals efficiently, offering a scalable solution. We propose using INRs to parameterize the watermark signal by coordinates, enabling continuous watermark generation at arbitrary positions and resolutions, overcoming the limitations of CNN-based methods for efficient embedding.

## 3  Methodology

Our approach is based on Implicit Neural Representations (INRs), the key idea of our method is to parametrize a template watermark using coordinates, with INRs serving as the rendering function for the watermark signal. A comprehensive architecture is presented in Figure 2, illustrating the key components of our framework. The approach is built upon three core modules as shown in the Figure: (a) a resolution-independent sampling strategy combined with hierarchical multi-scale coordinate embedding, ensuring consistency and robustness of watermark across varying image resolutions with a fixed and limited computational cost; (b) low-rank watermark injection based on Implicit Neural Representations, which reduces computational cost while achieving robust watermark embedding and (d) noise enhancement and decoding of the watermarked image.

### 3.1 Sampling and Embedding

**Resolution-Independent Coordinates Sampling.** For an image of size $(H, W)$, we normalize the pixel coordinates $(x, y)$ to the range $[-1, 1]$ as follows:

$$(x, y) \rightarrow \left( \frac{2x}{H} - 1, \frac{2y}{W} - 1 \right) \tag{1}$$

The normalized coordinate matrix obtained is denoted as $\psi$ in Figure 2 (a). To sample submatrices of arbitrary size from $\psi$, we use a fixed $r \times r$ coordinate grid $\mathcal{C}$, where $r$ is typically set to 128. The coordinates of the sampled submatrix are determined by the upper-left corner $(x_0, y_0)$, and $\mathcal{C}$ is defined as:

$$\mathcal{C} = \{(x_i, y_j) \mid x_i = x_0 + i \cdot \Delta_t, y_j = y_0 + j \cdot \Delta_t\} \tag{2}$$

where $i, j \in [0, r-1]$ and $\Delta_t$ is a randomly selected interval. The value of $\Delta_t$ controls the resolution of $\mathcal{C}$, allowing flexible extraction of regions at different scales while maintaining a fixed grid.

**Hierarchical Multi-Scale Coordinates Embedding.** Many prior works (Müller et al., 2022; Girish et al., 2023) have shown that using only coordinates as input leads to longer training times, loss of high-frequency details, and poor scalability for high-resolution signals. This is a challenge, as we aim to decode the watermark at arbitrary resolutions while maintaining robustness to cropping and scaling. Therefore, modeling multi-scale features of the watermark is crucial.

To address this issue, we propose the use of a set of feature grids with varying resolutions $L = \{L_i\}^n$ to represent the embedded features of the watermark template, where $n$ (default 4) denotes the number of feature grids. Each grid $L_i \in \mathbb{R}^{d \times 2^{i+4} \times 2^{i+4}}$ is a learnable parameterized matrix, with $d$ (default 32) representing the dimension of the features. The coordinates of the feature vectors in each grid are normalized to the range $[-1, 1]$. Given an input coordinate $(x, y)$, we identify the four nearest corner features in the matrix $L_i$, with the bottom-left and top-right corner features having coordinates $(x_{bl}^i, y_{bl}^i)$ and $(x_{tr}^i, y_{tr}^i)$, respectively. By applying bilinear interpolation, we can obtain the feature representation corresponding to any input coordinate in matrix $L_i(x, y)$:

$$L_i(x, y) = \begin{bmatrix} x_{tr}^i - x & x - x_{bl}^i \end{bmatrix} \begin{bmatrix} L_i(x_{bl}^i, y_{bl}^i) & L_i(x_{bl}^i, y_{tr}^i) \\ L_i(x_{tr}^i, y_{bl}^i) & L_i(x_{tr}^i, y_{tr}^i) \end{bmatrix} \begin{bmatrix} y_{tr}^i - y \\ y - y_{bl}^i \end{bmatrix} k \tag{3}$$

where $k = \frac{1}{(x_{tr}^i - x_{bl}^i)(y_{tr}^i - y_{bl}^i)}$. Additionally, to further enhance the ability of INR to represent high-frequency details, we apply Fourier feature mapping $\mathcal{F}$ to process the raw coordinates (Tancik et al., 2020b), obtaining the coordinate encoding $z_p$. Therefore, for any given coordinate $(x, y)$, we can obtain its corresponding feature representation $z_e \in \mathbb{R}^{d_e}$:

$$z_e = \Gamma(L_1(x, y) \odot L_2(x, y) \cdots L_n(x, y) \odot z_p) \tag{4}$$

where "$\odot$" denotes concatenation along the feature dimension and $\Gamma(*)$ is a linear layer used to project the features into the $d_e$-dimensional space (default 256).

**Message Embedding.** We use a four-layer MLP with fully connected layer, BatchNorm, and ReLU activation function as the Message Encoder $\mathcal{M}$. For a $t$-length binary message $m = \{0, 1\}^t$, we obtain the corresponding message feature $z_m = \mathcal{M}(m) \in \mathbb{R}^{d_m}$. $d_m$ defaults to 128.

### 3.2 Low-rank Watermark Injection based on Implicit Neural Representations

**INR Rendering.** As shown in Figure 2 (b), given a positional feature embedding $z_e$ and a watermark feature $z_m$, we use an INR to generate the watermark signal for the corresponding pixel location. This process is repeated for all pixel positions, and the results are reshaped to form the final watermark signal $W$. Then we randomly sample an $r \times r$ image block $I$ and obtain the watermarked image as $I' = I + \gamma W$, where $\gamma$ (default 0.02) controls embedding strength. A smaller $\gamma$ helps distribute the watermark evenly, enhancing visual quality and robustness.

**Low-rank Watermark Injection.** A simple INR-based approach is to concatenate two features and process them through an MLP to predict RGB values. However, prior research (Zadeh et al., 2017; Liu et al., 2018) has demonstrated that such direct concatenation leads to inadequate feature

interaction between modalities. Following TFN (Zadeh et al., 2017), we compute the Cartesian product of $z_e$ and $z_m$ to facilitate richer cross-modal feature interactions. This can be formulated as:

$$z = M \cdot \text{Vec}(\alpha\beta^\top) + bias \tag{5}$$

Where $\alpha = [z_e, 1] \in \mathbb{R}^n$, $\beta = [z_m, 1] \in \mathbb{R}^m$ and $\text{Vec}(*)$ the vectorization operator. The matrix $M \in \mathbb{R}^{h \times (n \times m)}$ is a learnable parameter, and $z$ is the final output feature. $h$ is the dimension of the output feature, default is 256.

However watermark injection occurs independently at each pixel location in our method. This results in significant computational and memory overhead. To mitigate this, we reformulate it (ignoring bias) as:

$$z = M \cdot \text{Vec}(I\alpha\beta^\top) = M(\beta \otimes I)\text{Vec}(\alpha) = M(\beta \otimes I)\alpha \tag{6}$$

We expand $M$ and $\beta \otimes I$ into a block matrix form as follows:

$$z = \begin{bmatrix} M_1 & M_2 & \cdots & M_m \end{bmatrix} \begin{bmatrix} \beta_1 I \\ \beta_2 I \\ \vdots \\ \beta_m I \end{bmatrix} \alpha = \sum_{i=1}^{m} \beta_i M_i \alpha \tag{7}$$

Where $M_i \in \mathbb{R}^{h \times n}$ can be viewed as a slice of a third-order tensor $P \in \mathbb{R}^{m \times h \times n}$, and $\sum_{i=1}^{m} \beta_i M_i$ can be interpreted as a weighted sum of the slices of $P$. Thus, our goal is to reduce the parameter count of the learnable tensor $P$.

Using Canonical Polyadic Decomposition (CPD) for low-rank approximation, we introduce three small learnable matrices $u \in \mathbb{R}^{m \times d}$, $v \in \mathbb{R}^{h \times d}$, and $w \in \mathbb{R}^{n \times d}$, where $d$ is the rank (defaults 32). Thus, any element in $P$ can be expressed as $p_{ijk} = \sum_{r=1}^{d} u_{ir} v_{jr} w_{kr}$. Then for any element of the output feature $z$ it can be expressed as:

$$\begin{aligned} z_j &= \sum_{k=1}^{n} \sum_{i=1}^{m} \beta_i p_{ijk} \alpha_k = \sum_{k=1}^{n} \sum_{i=1}^{m} \beta_i \sum_{r=1}^{d} u_{ir} v_{jr} w_{kr} \alpha_k \\ &= \sum_{k=1}^{n} \sum_{i=1}^{m} \sum_{r=1}^{d} \beta_i u_{ir} v_{jr} w_{kr} \alpha_k \end{aligned} \tag{8}$$

After simplification, we obtain: $z = v*((w^\top *\alpha)\circ(u^\top *\beta))$. Where "$*$" denotes matrix multiplication and "$\circ$" denotes element-wise multiplication. To preserve the rich semantic information contained in the features $\alpha$, we employ skip-connection to mitigate potential information loss caused by low-rank decomposition. Additionally, we stack 4 identical modules to ensure the robust injection of the watermark information. Figure 2 (c) illustrates our INR Block. It can be expressed as:

$$\alpha_i = \text{Relu}(\Psi(v_{i-1} * ((w_{i-1}^\top * \alpha_{i-1}) \circ (u_{i-1}^\top * \beta)))) + \alpha_{i-1} \tag{9}$$

Where $\Psi(*)$ denotes 1D batch normalization. A final FC layer maps features to RGB values, and after reshaping, the watermark $W \in \mathbb{R}^{3 \times r \times r}$ is obtained.

### 3.3 Noise Layer and Message Decoder

Figure 2 (d) illustrates the decoding process of our model. To enhance generalization, we introduce a composite noise layer to simulate real-world distortions. The Message Decoder then extracts the watermark from the distorted image.

**Noise Layer.** The noise layer consists of Rotation, Cropping, Translation, Scaling, Shearing, Dropout, Cropout, Color changes, JPEG compression, Gaussian filtering, and Gaussian noise. During training, a random noise type is applied to the watermarked image $I'$, producing $\hat{I}$.

**Message Decoder.** Our Message Decoder incorporates the SE Block (Hu et al., 2018) inspired by MBRS (Jia et al., 2021). It processes the input through four stacked blocks of identical structure. Each block consists of a ConvBNReLU layer with a kernel size of 3, followed by an SE Block and another convolutional layer. Finally, downsampling is performed using a convolutional layer with a kernel size of 4 and a stride of 2. The extracted features are pooled along the channel dimension, and a fully connected layer predicts the watermark information $\hat{m}$.

### 3.4 Loss Function

Our loss function comprises two components: the first aims to preserve the visual quality of the watermarked image $I^{'}$, while the second seeks to minimize the discrepancy between the extracted watermark $\hat{m}$ and the embedded watermark $m$. Both components are formulated using the mean squared error (MSE) loss. The total loss is given by:

$$\mathcal{L} = \lambda_1 \left\| I - \hat{I} \right\|_2 + \lambda_2 \left\| m - \hat{m} \right\|_2 \tag{10}$$

where $\lambda_1$ and $\lambda_2$ are hyperparameters that balance the trade-off between visual quality preservation and watermark extraction accuracy. By default, both are set to $1$.

## 4 Experiments

In this section, we conduct extensive experiments to evaluate the effectiveness of our proposed INR-based watermarking method. First, we describe the experimental setup. Then, we compare our method with previous SOTA models under low resolution. Subsequently, we evaluate our method across various resolutions against other large-image watermarking approaches, demonstrating its superior performance. Finally, ablation studies assess the contribution of proposed components.

### 4.1 Experimental Setting

**Implementation Details.** Our model is trained on the high-resolution DIV2K (Agustsson and Timofte, 2017) image dataset. For each training iteration, we randomly select an image from the dataset and apply a random scaling operation, where the scaling factor is chosen from the range $[0.06, 1]$. Following the scaling, we randomly crop a $128 \times 128$ image patch from the scaled image, which serves as the input $I$ to the model. For the watermark information $m$, we randomly generate a binary bit stream of length 30 in each iteration. For the sampling coordinates $\mathcal{C}$, we perform random sampling from a normalized coordinate grid $\psi$, using a fixed $128 \times 128$ grid size. We randomly generate training samples, with a training set size of $50,000$ samples. The model is trained using the AdamW (Loshchilov and Hutter, 2017) optimizer with a learning rate of $4 \times 10^{-4}$. The batch size is set to 32, and the training is conducted for 2000 epochs across two NVIDIA RTX 3090 24G GPUs.

**Metrics.** We evaluate our method using three metrics: Peak Signal-to-Noise Ratio(PSNR) for visual quality, Structural Similarity Index(SSIM) for structural similarity and Average Bit Accuracy (ACC) for average decoding accuracy. PSNR and SSIM assess image quality, while ACC measures watermark extraction robustness.

**Baseline.** We use HiDDeN (Zhu et al., 2018), StegaStamp (Tancik et al., 2020a), MBRS (Jia et al., 2021), DWSF (Guo et al., 2023), TrustMark (Bui et al., 2023) and RAIM$_{\text{ARK}}$ (Wang et al., 2024) as baselines. The first three methods are limited to low-resolution images, while others can work at different resolutions. To ensure a fair comparison, we re-train these models (excluding TrustMark, which does not provide a training script) with our proposed noise layer, as the strength of the noise layer significantly impacts their performance. Initially, we train our model on $128 \times 128$ images and compare them with the low-resolution baselines to demonstrate the effectiveness of our method. Subsequently, we relax the resolution constraint, allowing models to be trained on images with different sizes, and compare them with corresponding large-image watermarking models to validate the superiority of our approach.

### 4.2 Visual Quality

Table 1 presents the visual quality across different methods. Our method does not achieve the highest PSNR and SSIM values, but it consistently maintains a PSNR above 35. This result is expected, as our approach employs a template-based watermark, which does not leverage the content of the cover image for embedding. As a result, while the visual quality is slightly lower compared to methods that embed watermarks based on the cover image content, our method still achieves a PSNR above 35, ensuring it meets practical requirements for everyday use.

In addition, as shown in Figure 3, we also present the visualization results of our method at different resolutions. We observe that watermarks generated from different embedded messages exhibit

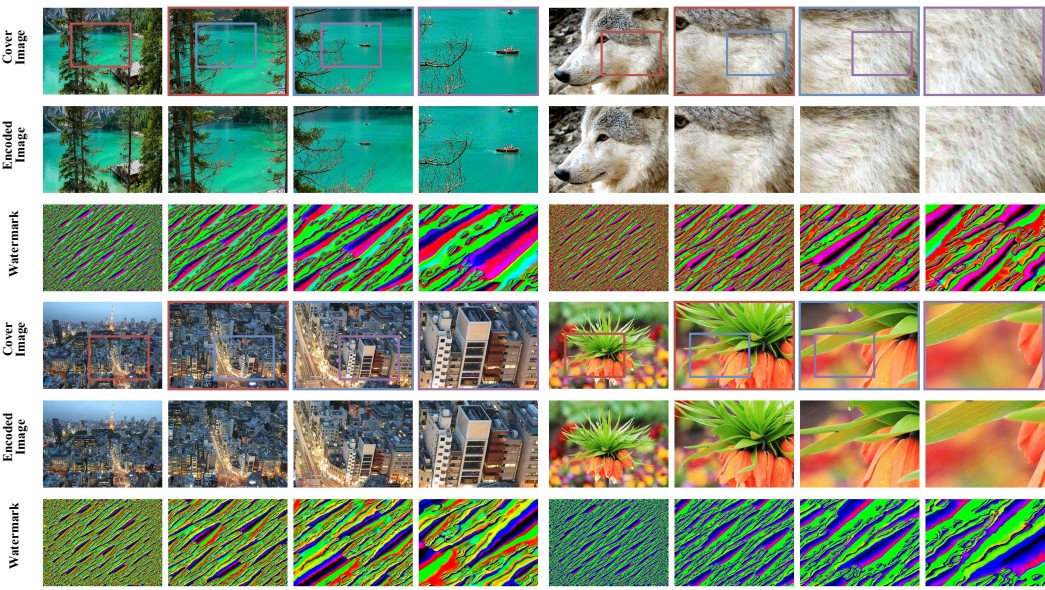

Figure 3: Visualization results at different resolutions. For each group, the first row presents cover images at different resolutions, starting from 2K size and progressively reduced by 75%. The second row displays the corresponding watermarked images, while the third row shows the embedded watermarks.

| | High Resolution | | | | Low Resolution | |
|---|---|---|---|---|---|---|
| | Proposed | DWSF | TrustMark | RAIM | HiDDeN | StegaStamp | MBRS |
| PSNR | 37.27 | 33.18 | 40.76 | 40.52 | 32.67 | 35.25 | 40.73 |
| SSIM | 0.9611 | 0.9269 | 0.9900 | 0.9898 | 0.8591 | 0.8864 | 0.9866 |

Table 1: Average visual quality of the different methods. For High Resolution methods, we measured at 2K size (except for DWSF). For Low Resolution methods, we measured at $128 \times 128$ size. For DWSF, it essentially embeds the watermark into many $128 \times 128$ blocks, so we measured the visual quality of the block.

highly similar overall structural patterns, with variations primarily reflected in color and fine details. Moreover, the watermark consistently maintains a similar striped pattern and distribution across various resolutions, indicating its structural consistency across different scales. This ensures that the watermark information can be successfully decoded from cropped patches at different resolutions.

### 4.3 Comparison with Previous Methods

We compare our method with previous SOTA models at a fixed resolution of $128 \times 128$ to evaluate its effectiveness. For a comprehensive evaluation, we test the decoding accuracy of our trained model using a variety of distortions: Identity, Gaussian Noise($std = 0.01$), Gaussian Filter ($\sigma = 2$), JPEG Compression ($Q = 50$), Dropout ($p = 0.5$), Rotation ($deg = 10$), Translation ($dis = 0.1$) and Color Transform($f = 0.1$). To ensure a fair comparison, we fix the PSNR at 35 for all watermarked images following the approach in the MBRS (Jia et al., 2021) method.

Table 2 presents the experimental results. Notably, despite employing INRs rather than traditional CNNs for information embedding, our proposed method achieves the best decoding accuracy under most distortion conditions. Specifically, the proposed method maintains extremely high accuracy under conditions such as Gaussian noise, Gaussian filtering, rotation, translation, and color perturbation. Even under more severe transformations like JPEG compression and dropout operations, the method demonstrates strong robustness, achieving accuracy rates of 93.36% and 95.13%, respectively.

| Method | Identity | Gaussian Noise $(std=0.01)$ | Gaussian Filter $(\sigma=2)$ | JPEG Compression $(Q=50)$ | Dropout $(0.5)$ | Rotation $(deg=10)$ | Translation $(dis=0.1)$ | Color $(f=0.1)$ |
|---|---|---|---|---|---|---|---|---|
| HiDDeN | 88.34 | 87.96 | 60.14 | 52.96 | 74.74 | 82.94 | 82.85 | 90.79 |
| StegaStamp | 92.16 | 91.72 | 90.78 | 84.42 | 77.53 | 87.19 | 88.46 | 91.21 |
| MBRS | 99.18 | 98.06 | 94.93 | **96.56** | **95.75** | 95.37 | 96.27 | 98.37 |
| DWSF | 90.17 | 89.76 | 89.40 | 87.33 | 76.47 | 86.56 | 65.90 | 78.36 |
| TrustMark | 87.65 | 83.72 | 82.60 | 75.22 | 76.58 | 49.50 | 62.13 | 87.25 |
| RAIM$_{ARK}$ | 78.67 | 78.67 | 78.67 | 54.33 | 57.67 | 77.24 | 74.58 | 77.67 |
| Proposed | **100** | **99.83** | **98.86** | 93.36 | 95.13 | **99.73** | **99.59** | **99.93** |

Table 2: Benchmark comparisons on robustness against different distortions. Where the mean value of Gaussian Noise is 0, the kernel size of Gaussian Filter is 3, and JPEG Compression is simulated using Kornia (Riba et al., 2020).

| Distortions | Model | $128 \times 128$ | | | $512 \times 512$ | | | $2048 \times 2048$ | | | $4096 \times 4096$ | | |
|---|---|---|---|---|---|---|---|---|---|---|---|---|---|
| | | DIV2K | COCO | FFHQ | DIV2K | COCO | FFHQ | DIV2K | COCO | FFHQ | DIV2K | COCO | FFHQ |
| Cropping | DWSF | 90.17 | 89.83 | 78.93 | 53.28 | 54.10 | 52.73 | 50.80 | 50.73 | 50.64 | 50.03 | 50.33 | 50.12 |
| | TrustMark | 87.65 | 92.15 | 96.10 | 49.55 | 49.45 | 50.23 | 49.77 | 49.58 | 49.41 | 49.50 | 54.57 | 47.25 |
| | RAIM$_{ARK}$ | 78.67 | 77.33 | 78.30 | 53.33 | 48.67 | 45.33 | 54.00 | 48.67 | 46.67 | 54.35 | 49.46 | 46.85 |
| | Proposed | **99.99** | **99.85** | **99.91** | **99.86** | **99.84** | **99.95** | **98.74** | **94.80** | **93.83** | **85.94** | **86.93** | **83.23** |
| Scaling | DWSF | 90.17 | 89.83 | 78.93 | 89.56 | 89.40 | 77.90 | 80.70 | 80.83 | 67.07 | 63.73 | 63.43 | 56.57 |
| | TrustMark | 87.65 | 92.15 | 96.10 | 79.15 | 85.90 | 89.43 | 76.07 | 86.25 | 88.88 | 79.51 | 82.26 | 86.77 |
| | RAIM$_{ARK}$ | 78.67 | 77.33 | 78.30 | 61.67 | 63.76 | 66.52 | 62.33 | 64.37 | 63.16 | 62.67 | 64.48 | 62.89 |
| | Proposed | **99.99** | **99.85** | **99.91** | **99.86** | **99.88** | **99.96** | **98.66** | **99.16** | **99.33** | **99.33** | **92.83** | **98.66** |

Table 3: Average decoding accuracy of different models for extreme cropping and scaling at different resolutions.

This robust performance strongly demonstrates that INR-based embedding is an effective and highly resilient mechanism for steganography.

## 4.4 Evaluation across Varying Image Resolutions

In this section, we investigate the performance of our method across different resolutions. We compare it with several SOTA watermarking methods capable of operating at various resolutions.

As modern applications increasingly involve high-resolution images such as digital media, medical imaging, and professional photography, scalable and reliable watermarking technology has become essential. For high-resolution images, two key challenges are commonly encountered during transmission: (1) High-ratio scaling, where the image is significantly reduced in size and (2) High-ratio random cropping, where only a small portion of the image is retained. These operations severely compromise the integrity of embedded watermarks, particularly when using traditional spatial domain or CNN-based methods.

Table 3 presents the experimental results. To evaluate the generalizability of our model, we test it not only on DIV2K but also on 200 separately sampled images from each of the COCO (Lin et al., 2014) and FFHQ (Karras et al., 2019) datasets. We embed watermarks at different resolutions and evaluate the performance of various methods under extreme cropping and scaling. Specifically, for cropping, we randomly extract $128 \times 128$ patches for decoding, while for scaling, we uniformly resize the images to $128 \times 128$ before decoding. To ensure a fair comparison, the PSNR is fixed at 35 for all watermarked images.

As can be seen, our method is far superior to the others. DWSF, RAIM$_{ARK}$ and TrustMark are all struggling to resist cropping attacks in large-image watermarking. Although these methods perform reasonably well at low resolutions ($128 \times 128$), they encounter issues in high-resolution scenarios. For instance, DWSF cannot withstand cropping at high resolutions because it is difficult to ensure that the cropped image block contains a complete embedding block. TrustMark, which embeds watermarks at high resolutions through interpolation, retains less watermark information after cropping, leading to extraction failure. On the contrary, on 2K images, our method requires only **0.4%** of the image area to maintain a **98%** decoding accuracy. Notably, this high level of robustness extends even to 4K resolution, where our method still achieves over 85% decoding accuracy using only a 128×128 patch, which corresponds to just 0.1% of the total image area. This superiority primarily stems from our coordinate sampling strategy and INR's capability to model continuous signals.

| Model | Proposed | DWSF | TrustMark | RAIM$_{ARK}$ |
|---|---|---|---|---|
| CPU | ✔ | ✘ | ✘ | ✘ |
| Embedding Rate | **4ms** | 74ms | 308ms | > 20min |

Table 4: Embedding rates for different methods.Where the second row represents whether the embedding is done using only the CPU or not, and the third row represents the average watermark embedding rate for a single image with a resolution of 2K. We use AMD Ryzen 7 7840HS for our CPU and NVIDIA RTX 3090 24G for our GPU for testing.

In addition, we test the performance of different distortions at different resolutions. As shown in Figure 4, our model achieves a performance of more than 90% at most resolutions, which is much better than other models. At the same time, we observe an interesting phenomenon: at low resolutions, the performance of Gaussian filtering and JPEG compression deteriorates. Both types of distortion are related to the image's frequency content, suggesting that the current method still has limitations.

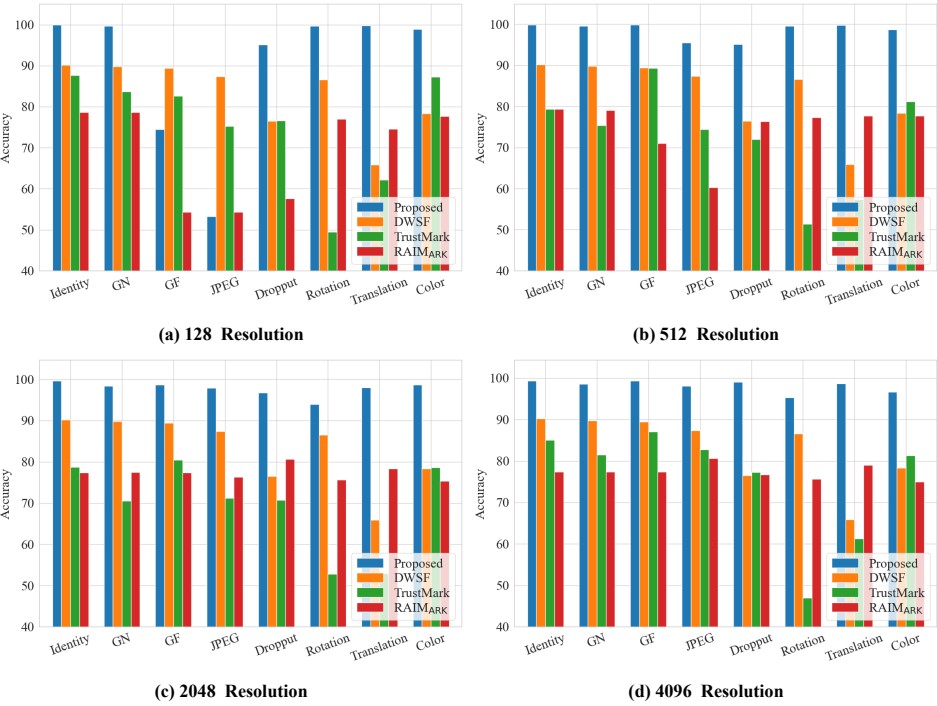

Figure 4: Average decoding accuracy of different models at different resolutions and distortions. Where GN stands for Gaussian Noise and GF stands for Gaussian Filter. To ensure fairness, the PSNR of all embedded images is standardized at 35.

## 4.5 Embedding Rate Comparison

In this subsection, we compare the computational efficiency between the different methods by the embedding rate of the watermark. Since our focus is on watermark embedding for large images, which often incur high computational costs due to their large resolution, the real-time performance of watermarking systems can be significantly impacted. As shown in Table 4, our method is significantly faster. This result primarily becomes it is based on template watermarking, allowing all watermarks to be pre-generated once the model is trained. In contrast, other methods require the network to process the carrier image during embedding, resulting in slower performance. Worth mentioning is that RAIM$_{ARK}$ requires INR encoding and watermark embedding fine-tuning for each image, making its embedding rate extremely slow.

|  | Coordinates Embedding (num of layers) | | | Low-rank Injection (rank) | | | |
|---|---|---|---|---|---|---|---|
|  | ✖ | 1 | 2 | 4 | ✖ | 8 | 32 | 64 |
| ACC | 90.91 | 91.66 | 93.86 | **95.17** | 93.77 | 84.69 | 95.17 | **95.76** |
| PSNR | 35.98 | 36.88 | 36.75 | **37.27** | 36.04 | 37.08 | **37.27** | 36.37 |
| SSIM | **0.9706** | 0.9590 | 0.9565 | 0.9611 | 0.9639 | **0.9656** | 0.9611 | 0.9550 |

Table 5: Model performance with different components. Here, "✖" indicates the absence of the corresponding component. For Coordinates Embedding, this means that only the Coordinate Fourier Mapper is used as an embedding feature. For Low-rank Injection, it signifies that features are concatenated and directly predicted by an MLP.

| Num of Bit | PSNR | SSIM | Avg ACC |
|---|---|---|---|
| 30 | 37.27 | 0.9656 | 95.76 |
| 50 | 35.59 | 0.9563 | 95.62 |
| 100 | 31.32 | 0.9123 | 88.26 |

Table 6: The relationship between visual quality and decoding accuracy at varying numbers of bits.

## 4.6 Ablation Study

In this subsection, we examine the effectiveness of each component of our proposed method, focusing on two main points: (1) the impact of the hierarchical multi-Scale coordinates embedding layers on model performance, and (2) the effectiveness of low-rank watermark injection.

Table 5 presents the results of our ablation study. We embed watermarks into 2K-resolution images and evaluate the average decoding accuracy after applying random attacks. The results show that removing multi-scale coordinate embedding significantly degrades performance, while increasing the number of embedding layers progressively improves it. For Low-rank Injection, replacing our design with feature concatenation followed by MLP prediction results in inferior performance. Moreover, a lower rank significantly degrades accuracy.

Additionally, we evaluated the watermarking capacity of our method. Table 6 illustrates the trade-off between watermark capacity, visual quality, and decoding robustness. As the embedded bit count increases from 30 to 100, the visual quality of images deteriorates, while accuracy remains at a high level. Even under 100 bits, the accuracy exceeds 88%. Since our method handles high-intensity cropping and scaling while maintaining high accuracy under current high payloads, it indicates that our embedding strategy inherently introduces significant redundancy. Therefore, exploring more efficient embedding patterns in the future to reduce redundancy and increase watermark capacity while maintaining robustness will be a highly meaningful research direction.

## 5 Conclusion

In this paper, we introduce a novel watermarking framework that leverages implicit neural representations and a resolution-independent coordinate sampling for efficient watermark embedding and extraction across images of high resolution. Unlike CNN-based methods that require processing entire images, our approach can embed watermark at the pixel level, enabling watermark embedding across different scales while avoiding excessive computational overhead and ensuring robustness against extreme cropping and scaling distortions. Additionally, we introduce a hierarchical multi-scale coordinate embedding and low-rank watermark injection to enhance model robustness. Experimental results show that our method outperforms existing approaches in both performance and computational efficiency. These findings highlight the potential of INR-based methods for high resolution watermarking solutions, offering valuable insights for future research on resolution-independent image watermarking.

## Acknowledgments and Disclosure of Funding

This work was supported in part by the Natural Science Foundation of China under Grant 62372203 and 62302186, in part by the Major Scientific and Technological Project of Shenzhen (202316021), in part by the National key research and development program of China(2022YFB2601802), in part by the Major Scientific and Technological Project of Hubei Province (2022BAA046, 2022BAA042).

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

# A Appendix

## A.1 Details of the Noise Layer

The combined noise layer is implemented using Kornia (Riba et al., 2020), incorporating various transformations such as Identity, Rotation, Cropping, Translation, Scaling, Shearing, Dropout, Cropout, Color Transformation, JPEG Compression, Gaussian Filtering, and Gaussian Noise. These transformations are applied as follows:

- **Rotation:** Random rotation within the angle range of $[-30°, 30°]$.

- **Cropping:** Randomly retains a scale of $[0.5, 1]$ of the original image.

- **Translation:** Generates random displacements of $[-0.1, 0.1]$ times the image's edge length along the x and y axes.

- **Scaling:** Random scaling within a factor of $[0.5, 1.2]$.

- **Shearing:** Random shear transformation with an angle range of $[-0.1, 0.1]$.

- **Dropout:** Randomly discards $[10\%, 30\%]$ of the pixels.

- **Cropout:** Randomly discards blocks of the image with a scale of $[0.05, 0.1]$.

- **Color Transformation:** Perturbs Brightness, Saturation, and Hue with intensities of $[-0.4, 0.4]$, $[-0.4, 0.4]$, and $[-0.1, 0.1]$, respectively.

- **JPEG Compression:** Applies random quality factors between $[50, 100]$.

- **Gaussian Filter:** Applies Gaussian filters with kernel sizes in the range of $[3, 8]$ and intensities of $[0.05, 0.1]$ with $\sigma \in [0.1, 2]$.

- **Gaussian Noise:** Adds Gaussian noise with a mean of $0$ and variance of $0.01$.

During training, a random noise type is selected and applied to perturb the watermarked image.

## A.2 The Role of Hyperparameter $\gamma$

In the section "Low-rank Watermark Injection based on Implicit Neural Representations", we introduced the $\gamma$ parameter to regulate the embedding strength of the watermark. Since our INR-based watermarking approach essentially functions as a template watermark, its pattern remains independent of the carrier image. As a result, the model naturally tends to generate sparse high-frequency textures to minimize visual loss, leading to significant spatial inefficiencies.

As illustrated in Figure 5, the left side shows the watermark generated without constraints. A substantial portion of the area contains null values, which not only wastes available space but also poses a critical issue—If the image is cropped to a region containing only null values, the essential watermark information may be lost entirely during transmission. To address this, we impose a constraint on the embedding strength, ensuring that the watermark is more uniformly distributed across the image.

Moreover, we empirically set $\gamma = 0.02$, as it maintains a PSNR of approximately 34 while preserving good visual quality, even for a completely randomized watermark template. If $\gamma$ is reduced further, the watermark's robustness deteriorates due to insufficient redundancy.

## A.3 TrustMark Settings in Baseline

Since TrustMark does not provide a training script, we rely on its pre-trained weights for testing. However, TrustMark's pre-trained watermarks are 100 bits long and are available in four open-source versions. We use the version with a 40-bit payload (BCH_SUPER), with the remaining 60 bits reserved for error correction and versioning. To ensure a relatively fair comparison, we modify the decoding process: after extracting the full 100-bit sequence, we first apply error correction to the 40 valid bits and discard the remaining 60 bits. The evaluation is then based solely on the decoding accuracy of the 40 valid bits.

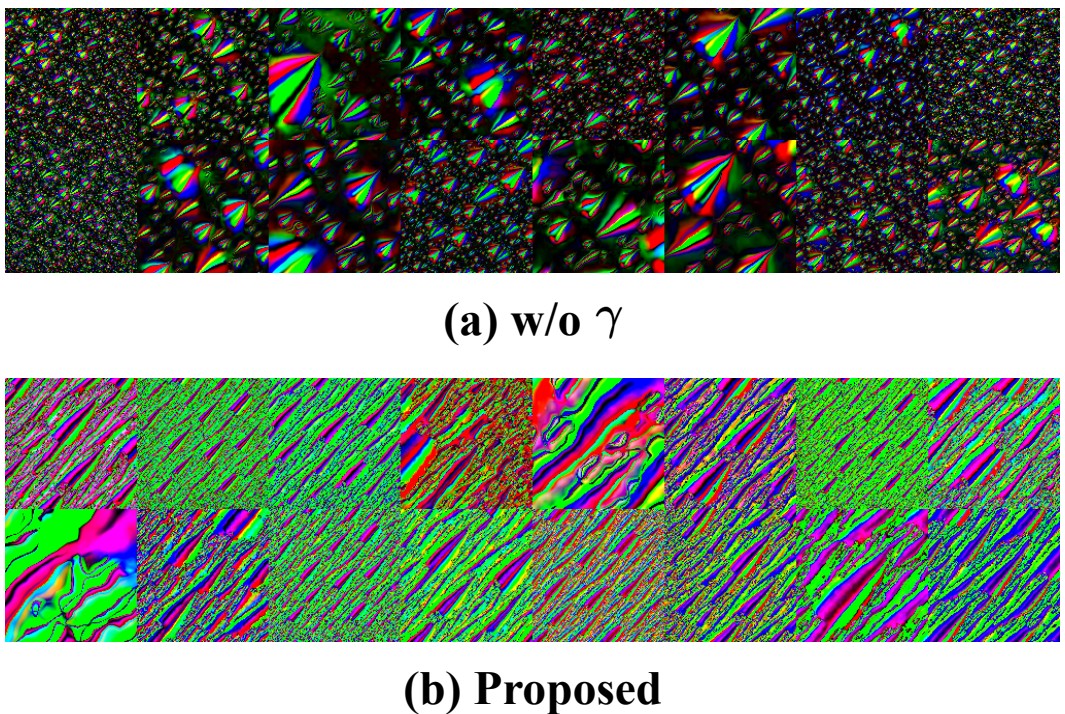

## (a) w/o $\gamma$

## (b) Proposed

Figure 5: Different watermark styles generated by the hyperparameter $\gamma$.

### A.4 More Results

In the main text, we present visualization results across different resolutions. Here, we provide additional experimental results, as shown in Figure 6. We randomly select various resolutions for embedding, where each group's first row represents the cover images, the second row shows the watermarked images, and the third row displays the watermarks. It can be observed that while the watermarks exhibit scale-dependent variations at different resolutions, they consistently retain a similar stripe-like structure and maintain considerable complexity at each resolution. This ensures the successful decoding of watermark information across varying image scales.

## B  Limitations and Future Work

Our approach shows promise but has some limitations. The visual quality of the watermarked images is lower than content-dependent methods, mainly due to INR's weaker feature representation compared to CNN. Additionally, INR fitting can be slow. Future work will aim to address these issues by combining CNN with INR to enhance both visual quality and embedding efficiency. We will also explore optimizing INR fitting to reduce training time and improve scalability. Moreover, how to better enhance watermark capacity is another issue that needs to be addressed.

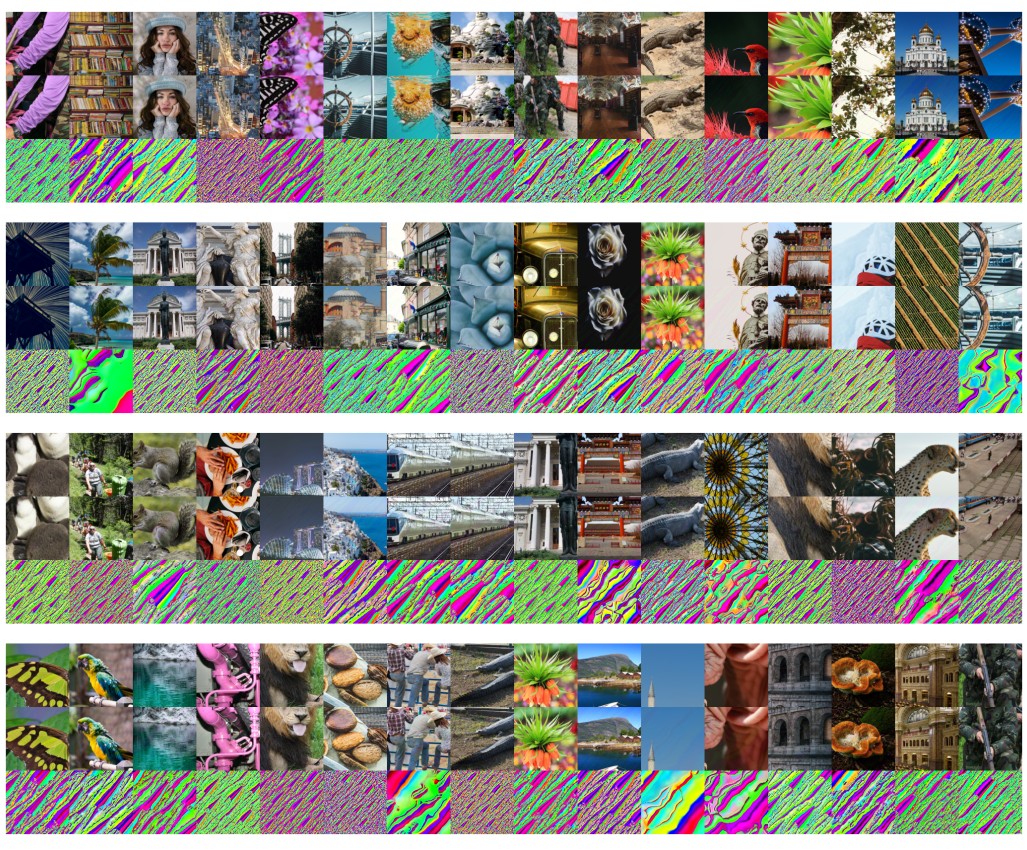

Figure 6: Visualization results at different resolutions.

