# OpenReview forum: "Ultra-high Resolution Watermarking Framework Resistant to Extreme Cropping and Scaling"
_NeurIPS.cc/2025/Conference — NeurIPS 2025 poster_

### Official Review · Reviewer_v2Wa · 2025-06-27

**Clarity:** 3
**Significance:** 3
**Originality:** 3
**Rating:** 5
**Confidence:** 4

**Summary:**

The manuscript introduces a novel watermarking framework designed for ultra-high resolution (UHR) images, leveraging implicit neural representations (INRs) to overcome the limitations of traditional deep neural network (DNN)-based watermarking techniques, which struggle with high computational and memory demands when applied to large-scale images. The proposed method utilizes a resolution-independent coordinate sampling mechanism to generate watermarks pixel-wise, ensuring efficient embedding with fixed computational resources. It incorporates hierarchical multi-scale coordinate embedding and a low-rank watermark injection strategy to enhance watermark quality and robustness against extreme distortions such as cropping and scaling. Experimental evaluations demonstrate that this approach outperforms existing methods in robustness and efficiency while maintaining acceptable image quality, making it a promising solution for protecting digital content in high-resolution contexts.

**Questions:**

1. Section 3.2 describes the low-rank watermark injection using tensor decomposition, but it does not sufficiently clarify the practical consequences for watermark quality and robustness. Specifically, how does the selection of rank influence the balance between computational efficiency and watermark performance? A quantitative example or analysis demonstrating this trade-off would greatly aid comprehension.
2. The resolution-independent coordinate sampling is highlighted as a key innovation, yet the manuscript lacks detail on how the sampling grid $\mathcal{C}$ adapts to varying image resolutions. Could the authors specify the process for determining the interval $\Delta_t$ and explain how it ensures consistent watermark generation across different scales?
3. What were the specific image dimensions used for the high-resolution and low-resolution categories in Table 1, and were these dimensions consistent across all methods within each category?
4. Were the models evaluated in Table 2 retrained for each of the resolutions tested, or was a single pretrained model applied across all resolutions without retraining?

**Ethical Concerns:**

["NO or VERY MINOR ethics concerns only"]

**Final Justification:**

My concern is addressed. I will keep my original positive score.

**Limitations:**

yes

**Paper Formatting Concerns:**

No significant formatting issues were observed.

**Quality:**

3

**Strengths And Weaknesses:**

Strengths
1. Innovative Application of Implicit Neural Representations (INRs) for Watermarking \
The manuscript proposes a pioneering approach by employing INRs to address the challenge of watermarking ultra-high resolution (UHR) images. Unlike traditional deep neural network (DNN)-based methods, which often falter under the computational and memory demands of large-scale images, this INR-based framework offers a resolution-independent solution. The use of coordinate sampling to generate pixel-wise watermarks is particularly noteworthy, as it ensures efficient embedding with fixed computational resources. This innovation fills a critical gap in the field and aligns well with the increasing demand for robust watermarking in high-resolution media applications.

2. Robustness to Extreme Distortions \
A standout feature of the proposed method is its exceptional resilience to extreme cropping and scaling distortions—common issues in the transmission of large images. The hierarchical multi-scale coordinate embedding preserves watermark integrity across varying scales, while the low-rank watermark injection enhances efficiency without sacrificing robustness. Experimental results are compelling, with watermark extraction accuracy exceeding 98% even when only 0.4% of a 2K image remains. This performance marks a significant improvement over existing techniques and underscores the practical value of the framework.

3. Comprehensive Experimental Validation \
The manuscript excels in its rigorous experimental evaluation, conducted across diverse datasets and multiple resolutions. The use of well-established metrics such as PSNR, SSIM, and decoding accuracy, coupled with comparisons to state-of-the-art methods, convincingly demonstrates the framework’s superiority in robustness and efficiency. The inclusion of visualization results and ablation studies further bolsters the credibility of the findings, providing readers with a clear understanding of the method’s strengths and limitations.

Weaknesses
1. Lack of Detailed Computational Complexity Analysis \
Despite touting computational efficiency as a major advantage, the manuscript falls short in providing a detailed analysis of the method’s computational complexity or memory requirements. For UHR images, these factors are paramount, and the absence of quantitative data, such as time complexity or memory footprint, hinders a full evaluation of the method’s feasibility. Including a comparative analysis of computational costs against baseline methods would have strengthened the manuscript’s claims and overall rigor.

2. Insufficient Clarity in Explaining Hierarchical Multi-Scale Coordinate Embedding \
The explanation of the hierarchical multi-scale coordinate embedding is the cornerstone of the proposed method, but it lacks clarity and accessibility. The technical description is dense and may prove difficult for readers without prior expertise in INRs or multi-scale techniques to grasp. The addition of illustrative diagrams or a step-by-step breakdown of the process would have significantly enhanced comprehension and made this key contribution more transparent to a broader audience.

---

> ### Author Rebuttal · Authors · 2025-07-30
>
> Thank you for your high praise of our work! We apologize for any confusion that may have been caused by unclear wording in the main text. We will make the necessary corrections in the camera-ready version.
>
> # Q1: Lack of detailed computational complexity analysis (shown in Weaknesses)
>
> We have already provided the inference time in Table 4. We mainly claim that we can train images of any size with fixed and limited computing resources. To this end, we provide the GPU usage (GB) per sample during training (Adam optim) for different methods:
>
> |          |    Proposed    | HiDDeN |   StegaStamp   | MBRS | DWSF | $\mathrm{RAIM_{ARK}}$ |
> | :-------: | :------------: | :----: | :------------: | :--: | :--: | :---------------------: |
> |  128x128  |      0.51      |  0.18  | **0.07** | 0.47 | 1.8 |          1.54          |
> |  512x512  | **0.51** |  5.04  |      0.75      | 4.73 |  -  |          11.93          |
> | 1024x1024 | **0.51** |  7.42  |     2.628     | 17.4 |  -  |          > 24          |
> | 2048x2048 | **0.51** |  >24  |     10.51     | >24 |  -  |          > 24          |
>
> It can be seen that the memory usage of our method does not increase with the image size.
>
> # Q2: The selection of rank influence the balance between computational efficiency and watermark performance
>
> In Section 4.6, we have already shown the impact of different ranks on visual quality and decoding accuracy in the ablation experiment.
>
> Here, we focus on the GPU memory consumption of different ranks with the corresponding number of INR parameters in training single samples, and we also provide the GPU memory consumption without using low-rank watermarking for watermark injection:
>
> |                    | w/o low-rank |  r=8  | r=32 | r=64 | r=128 | r=256 |
> | :-----------------: | :----------: | :---: | :---: | :--: | :---: | :---: |
> | GPU consumption(GB) |     > 24     | 0.49 | 0.51 | 0.53 | 0.60 | 0.71 |
> |   INR parameters   |    34.6M    | 22.6k | 84.2k | 166k | 330k | 659k |
>
> If we directly use formula (5) from the paper, single sample training would exceed 24GB memory. Since we use RTX 3090 with 24GB, this makes training impossible to complete.
>
> # Q3: Interval selection process and consistency guarantee
>
> For simplicity in the main text, we refer to this as a "randomly selected interval $\Delta_t$". However, in our actual implementation, $\Delta_t$ is `automatically determined` based on the selected region.
>
> We first randomly sample the top-left coordinates $(x,y)$ from the **"normalised coordinate matrix"** (shown in section 3.1) and randomly sample the submatrix side length $s\in[0,2]$. The code is as follows:
> ```python
>     def gen_start_coords():
>         side_length = np.random.uniform(0, 2)
>         start_x = np.random.uniform(-1, 1 - side_length)
>         start_y = np.random.uniform(-1, 1 - side_length)
>         return (start_x, start_y), side_length
> ```
>
> Then, we generate the subcoordinate matrix using the fixed grid parameter $r$. The code is as follows:
>
> ```python
>     x = torch.linspace(start_x, start_x + side_length, r)
>     y = torch.linspace(start_y, start_y + side_length, r)
>     xv, yv = torch.meshgrid(x, y, indexing='ij')
>     coordinates = torch.stack([xv, yv], dim=-1)
> ```
>
> As can be seen, as long as we sample a matrix of any size from the **"normalised coordinate matrix"**, we can determine a corresponding $\Delta_t$.
>
> Consistency is ensured by the intrinsic properties of INRs, which learn a continuous coordinate-to-RGB mapping. Since each coordinate maps to a unique RGB value and coordinates are processed independently during training, the `same coordinate will always yield the same RGB value` regardless of the sampling submatrix's size or location, thereby maintaining consistency across scales.
>
> # Q4: Specific image dimensions used for the high-resolution and low-resolution categories in Table 1
>
> For High Resolution, we measured at 2K size (except for DWSF), and for Low Resolution, we measured at 128x128 size. For DWSF, it essentially embeds the watermark into many 128x128 blocks, so we measured the visual quality of the block.
>
> We understand your concern that visual quality may vary across different resolutions, so we have also measured the visual quality of our method at different resolutions:
>
> |      | 128x128 | 512x52 | 1024x1024 | 2048x2048 |
> | :--: | :-----: | :----: | :-------: | :-------: |
> | PSNR |  37.46  | 37.56 |   37.34   |   37.27   |
> | SSIM | 0.9621 | 0.9632 |  0.9613  |  0.9611  |
>
> Our method maintains consistent visual quality at different resolutions.
>
> # Q5: Whether a single pretrained model is applicable across all resolutions without retraining
>
> As mentioned in Section 4.3, in order to make a fair comparison with the low-resolution model, Table 2 shows the results of training at a fixed resolution of 128×128 only.
>
> Table 3 and Figure 4 shows the performance of our method at varying image resolutions. It is trained only at 2K resolution and tested at different resolutions.
>
> We will clarify this point in the camera-ready version.

---

> > ### Comment · Reviewer_v2Wa · 2025-08-05
> >
> > My concerns are addressed. I will keep my original score.

---

> > > ### Author Response · Authors · 2025-08-06
> > >
> > > Dear Reviewer v2Wa,
> > >
> > > Thank you for your message and for taking the time to review our rebuttal. We are very happy to know that your concerns have been addressed.
> > >
> > > We would also like to express our sincere gratitude for your positive and supportive assessment throughout the review process. Your encouragement is highly valued by us.
> > >
> > > Sincerely,
> > >
> > > The Authors of Submission 15851

---

### Official Review · Reviewer_jFV8 · 2025-06-27

**Clarity:** 3
**Significance:** 3
**Originality:** 3
**Rating:** 5
**Confidence:** 4

**Summary:**

The work proposes watermarking of ultra-high resolution images by representing said watermarks as implicit neural representations. The goal is to make the watermarks resistant to extreme cropping and scaling. The authors propose to forego CNN modules and use a multiscale INR approach to encode watermarks in cover images, thus achieving their goal of robustness to cropping and scaling. Compared to SOTA watermarking methods, the proposed approach achieves higher visual quality and comparable decoding performance under different noise settings.

**Questions:**

See the weaknesses above. Besides that, It may not be the scope of the current work, but have the authors performed tests with printed images? From my understanding, this is not the focus of the work, although the authors also employ noise layers, used by StegaStamp [2] in their printer-proof method. Thus it would be interesting to have an idea whether the proposed method can be adapted to such media, and how straightforward would this adaption be.

**Ethical Concerns:**

["NO or VERY MINOR ethics concerns only"]

**Final Justification:**

Based on the other reviews and authors' answers, I believe that most concerns were answered adequately (at least my concerns were). I see no reason to not raise my rating. Good luck and excellent work to the authors!

**Limitations:**

Yes, the authors have adequately adressed the limitations.

**Paper Formatting Concerns:**

Not major, but Table captions should come above the content, as per NeurIPS template.

**Quality:**

3

**Strengths And Weaknesses:**

# Strengths
The work is well motivated and presents a clear goal. The idea of parameterizing the watermark using INRs adds flexibility and saves computational resources when using high-resolution images. I believe that it may impact in training time, however, I'm not really sure by how much. The idea of encoding images as INRs is fairly new and the application is of paramount importance to provide an authenticity stamp against deep-fake methods, even though the authors don't frame their work this way.

Furthermore, I appreciate that the authors submitted source-code for the review process, as it helps the reader to check for implementation details of the method.

# Weaknesses
I missed experiments with varying message sizes. The authors focus on 30 bit messages, following HiDDeN [1], which is significantly less than StegaStamp [2], and RiemStega [3], both more recent methods. Furthermore, [2,3,4] perform tests with varying message sizes, including 30, 50, 100, 150 and 200 bits, which are missing in the current work. This is important to assess the balance between image quality and method capacity to decode the watermark.

As I mentioned in the strengths, I appreciate that source-code was submitted, although it is missing instructions regarding package and versions, in addition to local packages used (FastTools).

Furthermore, there is no mention of training time of the proposed method. Since there is a major architectural change by swapping CNNs for INRs (I know the changes are much larger than that, bear with me), the reader needs to know the impact regarding training time. Typically INRs are not particularly lightweight on computational resource usage. which is reinforced by the authors' statement on Sec. A.6.

Finally, a more thorough hardware specification should be made. What is the memory ammount of the GPUs? The authors only list the model (RTX 3090), without memory and processor details, which may differ depending on the manufacturer.

# Minor comments
- Typo on abstract: "...with only 0. 4% of the image..." => extra space between point and digit.
- In Sec. 3, the first paragraph, where the approach is described. The authors mention three core modules and mark them as (a), (b), and (d).
- Table captions should come above the content, as per NeurIPS template.
- On the appendix, Figure 5. The subcaptions are really large compared to the remaining text

# References
[1] Zhu, J., Kaplan, R., Johnson, J., & Fei-Fei, L. (2018, September). HiDDeN: Hiding Data with Deep Networks. Proceedings of the European Conference on Computer Vision (ECCV).

[2] Tancik, M., Mildenhall, B., & Ng, R. (2020, June). StegaStamp: Invisible Hyperlinks in Physical Photographs. Proceedings of the IEEE/CVF Conference on Computer Vision and Pattern Recognition (CVPR).

[3] Cruz, A., Schardong, G., Schirmer, L., Marcos, J., Shadmand, F., & Gonçalves, N. (2025). RiemStega: Covariance-Based Loss for Print-Proof Transmission of Data in Images. 2025 IEEE/CVF Winter Conference on Applications of Computer Vision (WACV), 7572–7581. https://doi.org/10.1109/WACV61041.2025.00736

[4] Bui, T., Agarwal, S., & Collomosse, J. (2023). TrustMark: Universal Watermarking for Arbitrary Resolution Images (arXiv:2311.18297). arXiv. https://doi.org/10.48550/arXiv.2311.18297

---

> ### Author Rebuttal · Authors · 2025-07-30
>
> Thank you for your appreciation of our work! We will answer your questions one by one.
>
> # Q1: Capacity of framework
>
> Fixing the message length at 30 bits is a `common practice`. It is widely adopted in both low-resolution studies such as HiDDeN[1] (**2018**), MBRS[2] (**2021**), and recent high-resolution approaches like DWSF[3] (**2023**) and $\mathrm{RAIM_{ARK}}$[4] (**2024**).
>
> We present the relationship between visual quality and decoding accuracy for models trained with message payloads of 30, 50, and 100 bits:
>
> | num of bit | PSNR |  SSIM  | Avg ACC |
> | :--------: | :---: | :----: | :-----: |
> |     30     | 37.27 | 0.9656 |  95.76  |
> |     50     | 35.59 | 0.9563 |  95.62  |
> |    100    | 31.32 | 0.9123 |  88.26  |
>
> In addition, We find that convergence is quite difficult when exceeding 100 bits (the convergence rate is slow), and the visual quality deteriorate significantly.
>
> This is reasonable. In order to accurately extract watermarks at any size, it is inevitable that a large amount of redundant information will be embedded. Therefore, how to further improve capacity based on existing methods remains a topic worthy of further exploration.
>
> # Q2: Missing of local packages and instructions
>
> We apologize, but since FastTools is an internal package that contains a large amount of private information (such as Hugging Face user tokens and passwords), we have not yet organised it. We will organise clean code and clear instructions after the camera-ready version is complete and open source it.
>
> # Q3: Compare of training time
>
> Here is a comparison of training times using one RTX 3090 24G (32 batch):
>
> | Method | Proposed | HiDDeN | StegaStamp | MBRS | DWSF | TrustMark | $\mathrm{RAIM_{ARK}}$ |
> | :----: | :------: | :----: | :--------: | :--: | :--: | :-------: | :---------------------: |
> | Times |   26h   |  23h  |    19h    | 14h | 39h |     -     |    >20min per image    |
>
> As can be seen, CNN-based methods have a clear advantage in terms of training time, but we are still significantly faster than DWSF in training. Furthermore, $\mathrm{RAIM_{ARK}}$ requires fine-tuning for each image, which is unacceptable in large-scale production.
>
> # Q4: Hardware specification
>
> We provide detailed specifications for hardware and software.
>
> |        |                                          |
> | :----: | :---------------------------------------: |
> |  GPU  |        NVIDIA GeForce RTX 3090 24G        |
> |  CPU  | Intel(R) Xeon(R) Platinum 8269CY @ 2.5GHz |
> | Memery |  Samsung M393A4K40CB2-CTD DDR4 32G x 12  |
> |   OS   |            Ubuntu 20.04.1 LTS            |
> |  CUDA  |                   12.8                   |
>
> # Q5: Printing images distortion adaptation
>
> StegaStamp is able to cope with printing image distortion mainly because it implements noise layers that simulate this process. Therefore, we only need to replace our noise layer with theirs, which enables adaptation to printing image distortion.
>
> Additionally, our designed distortion layers already encompass their design to a certain extent. Therefore, we directly present our experimental results.
>
> we randomly select 50 2K images (PSNR at 35 dB), print them on A4 paper using the **HP Color Laser MFP 178nw** printer, and capture them using a **Redmi K20 Pro** at a distance of 20 cm from the image. The average accuracy rate we achieved is `91.2%`.
>
> # Q6: Format problem
>
> Thank you for your patience and attention to detail. We will correct these formatting errors in the camera-ready version.
>
> # Reference
>
> [1] Zhu J, Kaplan R, Johnson J, et al. Hidden: Hiding data with deep networks[C]//Proceedings of the European conference on computer vision (ECCV). 2018: 657-672.
>
> [2] Jia Z, Fang H, Zhang W. Mbrs: Enhancing robustness of dnn-based watermarking by mini-batch of real and simulated jpeg compression[C]//Proceedings of the 29th ACM international conference on multimedia. 2021: 41-49.
>
> [3] H. Guo, Q. Zhang, J. Luo, F. Guo, W. Zhang, X. Su, and M. Li, “Practical deep dispersed watermarking with synchronization and fusion,” in Proceedings of the 31st ACM International Conference on Multimedia, 2023, pp. 7922–7932.
>
> [4] Y. Wang, X. Zhu, G. Ye, S. Zhang, and X. Wei, “Achieving resolution-agnostic dnn-based image watermarking: A novel perspective of implicit neural representation,” in Proceedings of the 32nd ACM International Conference on Multimedia, 2024, pp. 10 354–10 362.

---

> > ### Comment · Reviewer_jFV8 · 2025-08-03
> > **Thank you for your efforts.**
> >
> > I would like to thank the authors for their efforts.
> >
> > >In addition, We find that convergence is quite difficult when exceeding 100 bits (the convergence rate is slow), and the visual quality deteriorate significantly.
> >
> > That was exactly my point when asking for these aditional experiments. These methods are a trade-off between image quality, decoding capacity and information density. I ask that the authors add these results somewhere on the text (main paper or supp. mat.), with sample images with each model size.
> >
> > I would also like to thank the authors for the printer-proof experiments and ask that these are added to the text, even if they are preliminary results.
> >
> > Finally, add the training-time and hardware configurations to the main text, please.
> >
> > Congratulations on a nice work! And good luck.

---

> > > ### Author Response · Authors · 2025-08-06
> > >
> > > Dear Reviewer jFV8,
> > >
> > > Thank you for your valuable and constructive review. We sincerely appreciate your guidance on improving our paper and your words of encouragement. We confirm that we will incorporate all the requested additions into our final manuscript.
> > >
> > > We are very grateful for your positive assessment and support.
> > >
> > > Sincerely,
> > >
> > > The Authors of Submission 15851

---

### Official Review · Reviewer_RwW9 · 2025-06-29

**Clarity:** 2
**Significance:** 3
**Originality:** 2
**Rating:** 4
**Confidence:** 4

**Summary:**

This paper presents an INR-based ultra-high resolution watermarking framework. Using resolution-independent coordinate sampling, hierarchical multi-scale embedding, and low-rank injection, it enables robust watermark extraction under extreme cropping/scaling, outperforming baselines in both robustness and efficiency with high image quality.

**Questions:**

See the weaknesses

**Ethical Concerns:**

["NO or VERY MINOR ethics concerns only"]

**Final Justification:**

I have read the rebuttal and other reviews, and my concerns are almost well addressed. Given the premise that the missing certain details and format issues will be solved, I raise my score to "4: Borderline accept".

**Limitations:**

yes

**Paper Formatting Concerns:**

The paper's citation format like "(1; 2; 3)", seems incorrect.

**Quality:**

2

**Strengths And Weaknesses:**

Paper Strengths:
1. This paper applies INRs to ultra-high resolution watermarking, offering a good perspective and breaking fixed resolution constraints via resolution-independent coordinate sampling for efficient watermark generation across scales.
2. Technical designs including hierarchical embedding and low-rank injection, achieve high performance under extreme cropping, outperforming baselines and enabling faster embedding speed.


Paper Weaknesses:
1. Uses template-based watermarks without leveraging cover image content, resulting in slightly lower visual quality, restricting applicability in scenarios with high quality requirements (e.g., professional photography). The paper could benefit from presenting more failure cases and discussions.
2. Experiments focus on single distortion types (e.g., cropping, scaling) and lack systematic evaluation under mixed distortions (e.g., extreme cropping + scaling + noise), leaving real-world transmission robustness unconfirmed.
3. Generated watermarks are mostly stripe-like; templated features may reduce concealment and make them vulnerable to targeted attacks.
4. Experiments use fixed-length (30-bit) watermarks, with no mention of adaptability to variable-length messages, limiting the flexibility.
5. Curious cite formats like "(1; 2; 3)", and some typos (e.g., line 18 in Abstract, "0. 4%" with extra space)

Overall, I appreciate the authors' efforts in presenting an ultra-high resolution watermarking framework. Considering the strengths along with the weaknesses in details, analysis and experiments, I lean towards "3: Borderline reject" at this stage.

---

> ### Author Rebuttal · Authors · 2025-07-30
>
> Thank you for your appreciation of our approach! Our main `contribution` lies in proposing a completely new paradigm to break through the inherent resolution problem. We apologise for any confusion caused by our oversight of certain details, and we will address each of your questions below.
>
> # Q1: Discussion about application scenarios due to template-based watermarking approach
> Template-based watermark selection involves a `trade-off`. The content-agnostic nature of the method is a double-edged sword. While it doesn't adapt to the image content, this very independence is what enables its offline generation and mass reusability, making it uniquely suited for scalable platforms like large image libraries, news agencies, content distribution platforms and so on.
>
> Moreover, we have provided many `examples` of embeddings at different sizes in Figure 3 and Figure 7. As shown in Figure 3 (first row) shows that smaller image sizes with simpler textures exhibit more obvious visual distortion.
>
> At the same time, we also explored how to utilise cover image content under the INR architecture, but encountered challenges. Our experimental methods are as follows:
>
> > (1) The RGB values of the original image and the feature $z_e$ in Section 3.1 are concatenated and fed into INR.
>
> > (2) Pre-computing the texture richness of each pixel using the Laplacian operator and normalising it to [0,1] as $\gamma$ (shown in Section 3.2). This approach effectively controls embedding only in high-frequency regions.
>
> This is result at DIV2K:
>
> | Methods | PSNR |  SSIM  | Avg ACC |
> | :------: | :---: | :----: | :-----: |
> | Proposed | 37.27 | 0.9611 |  95.17  |
> |   (1)   | 37.31 | 0.9632 |  94.67  |
> |   (2)   | 40.16 | 0.9811 |  80.41  |
>
> For method (1), the INR model `selectively ignores` raw RGB values, which makes sense since isolated pixels do not convey meaningful semantic information on their own.
>
> While using the Laplacian operator to constrain embedding to high-frequency regions improves visual quality, it significantly reduces decoding accuracy. This occurs because the Laplacian constraint ($\gamma$) and the predicted watermark residual are independent variables, making decoding accuracy unreliable.
>
>
> # Q2: Robustness of mixed distortions
>
> For clarity, we have not included the performance of combination distortion in the main text.
>
> Here is the `“noise+scaling+cropping"` results. The results represent the average accuracy rate of embedding on a 2K image(DIV2K datasets), first performing different distortion attacks, then scaling to different resolutions, and finally randomly cropping a 128x128 patch for decoding. For fair comparison, PSNR is unified to 35.
>
> |         128x128         |    Identity    |       GN       |       GF       |      JPEG      |     Dropout     |    Rotation    |   Translation   |      Color      |
> | :---------------------: | :-------------: | :-------------: | :-------------: | :-------------: | :-------------: | :-------------: | :-------------: | :-------------: |
> |         propsed         | **99.99** | **99.93** | **99.93** | **99.97** | **99.99** | **99.90** | **99.99** | **97.40** |
> |          DWSF          |      90.17      |      89.76      |      89.4      |      87.36      |      76.4      |      85.56      |      65.9      |      78.36      |
> |        TrustMark        |      87.65      |      83.72      |      82.6      |      75.22      |      75.7      |      49.5      |      62.13      |      87.25      |
> | $\mathrm{RAIM_{ARK}}$ |      78.67      |      72.43      |      54.24      |      56.43      |      64.23      |      57.42      |      54.63      |      60.33      |
>
> |         512x512         |    Identity    |       GN       |       GF       |      JPEG      |     Dropout     |    Rotation    |   Translation   |      Color      |
> | :---------------------: | :-------------: | :-------------: | :-------------: | :-------------: | :-------------: | :-------------: | :-------------: | :-------------: |
> |         propsed         | **99.86** | **99.89** | **99.57** | **98.90** | **99.85** | **99.81** | **99.81** | **98.07** |
> |          DWSF          |      53.28      |      54.36      |      55.39      |      52.36      |      51.76      |      54.1      |      52.93      |      53.33      |
> |        TrustMark        |      79.55      |      70.39      |      74.72      |      61.55      |      72.3      |      71.12      |      59.07      |      70.65      |
> | $\mathrm{RAIM_{ARK}}$ |      53.24      |      53.1      |      52.53      |      52.54      |      54.21      |      50.53      |      49.2      |      50.54      |
>
> |        2048x2048        |    Identity    |       GN       |       GF       |      JPEG      |     Dropout     |    Rotation    |   Translation   |      Color      |
> | :---------------------: | :-------------: | :-------------: | :-------------: | :-------------: | :-------------: | :-------------: | :-------------: | :-------------: |
> |         propsed         | **98.74** | **98.40** | **77.27** | **62.97** | **98.03** | **97.30** | **93.30** | **94.23** |
> |          DWSF          |      50.83      |      50.93      |      50.7      |      50.5      |      49.93      |      50.92      |      50.21      |      50.87      |
> |        TrustMark        |      52.5      |      51.43      |      51.25      |      50.75      |      50.25      |      50.75      |      50.5      |      51.25      |
> | $\mathrm{RAIM_{ARK}}$ |      54.64      |      54.21      |      51.23      |      53.26      |      54.16      |      53.53      |      50.76      |      54.32      |
>
> |        4096x4096        |    Identity    |       GN       |       GF       |      JPEG      |     Dropout     |    Rotation    |   Translation   |      Color      |
> | :---------------------: | :-------------: | :-------------: | :-------------: | :-------------: | :-------------: | :-------------: | :-------------: | :-------------: |
> |         propsed         | **85.95** | **87.97** | **58.07** | **52.10** | **87.40** | **77.87** | **74.97** | **76.70** |
> |          DWSF          |      50.02      |      49.9      |      50.73      |      50.16      |      49.16      |      50.73      |      50.12      |      50.16      |
> |        TrustMark        |53.99|51.34|51.25|50.75 |      50.25      |      50.75      |      50.54      |      48.99      |
> | $\mathrm{RAIM_{ARK}}$ |54.14 |54.25|53.24 | 50.25 |54.34| 54.05 |53.12 | 52.61 |
>
> Other methods cannot withstand resolutions higher than 512x512, while our method performs well under most combinations of noise.
>
> # Q3: Discussion about stripe-like watermark pattern
>
> We acknowledge this limitation may reduce concealment and make them vulnerable to targeted attacks and believe it represents a significant area for future improvement. During our early explorations, we found that using SIREN [1] as the base INR block or employing different parameter initialization methods for the Hierarchical Multi-Scale Coordinates Embedding resulted in `different watermark textures` (e.g., fractal-like effects). However, most of these approaches led to training failures where visual quality improved but decoding accuracy remained at 0.5.
>
> Consequently, we chose the ReLU activation function for INR and default parameter initialization as the most stable option. This finding suggests that model structure and initialization methods significantly influence watermark texture, making this a promising direction for future research.
>
> # Q4: Discussion about variable-length messages
> We designed the message following `mainstream methods` such as HiDDeN[2] (2018), MBRS[3] (2021), DWSF[4] (2023), and $\mathrm{RAIM_{ARK}}$[5] (2024), all of which are fixed at 30 bits. As our primary focus is on proposing a new paradigm to effectively embed watermarks in high-resolution images, we adopt consistent designs with mainstream methods in other aspects.
>
>
> To the best of our knowledge, only training-free diffusion-based methods [6,7] can dynamically change embedding bit counts during inference. Other trained image watermarking methods cannot modify number of bits at inference time.
>
> Nevertheless, we can approximate dynamic bit changes through redundant embedding. For example, if trained with 6 bits but only 3 bits are needed (e.g., "010"), we embed "010|010" instead. This method can improve robustness through voting.
>
> # Q5: Format problem
>
> Thank you for your careful review and attention to detail. We will fix these formatting issues in the camera-ready version.
>
> # Reference
>
> [1] Sitzmann V, Martel J, Bergman A, et al. Implicit neural representations with periodic activation functions[J]. Advances in neural information processing systems, 2020, 33: 7462-7473.
>
> [2] Zhu J, Kaplan R, Johnson J, et al. Hidden: Hiding data with deep networks[C]//Proceedings of the European conference on computer vision (ECCV). 2018: 657-672.
>
> [3] Jia Z, Fang H, Zhang W. Mbrs: Enhancing robustness of dnn-based watermarking by mini-batch of real and simulated jpeg compression[C]//Proceedings of the 29th ACM international conference on multimedia. 2021: 41-49.
>
> [4] H. Guo, Q. Zhang, J. Luo, F. Guo, W. Zhang, X. Su, and M. Li, “Practical deep dispersed watermarking with synchronization and fusion,” in Proceedings of the 31st ACM International Conference on Multimedia, 2023, pp. 7922–7932.
>
> [5] Y. Wang, X. Zhu, G. Ye, S. Zhang, and X. Wei, “Achieving resolution-agnostic dnn-based image watermarking: A novel perspective of implicit neural representation,” in Proceedings of the 32nd ACM International Conference on Multimedia, 2024, pp. 10 354–10 362.
>
> [6] Wen Y, Kirchenbauer J, Geiping J, et al. Tree-ring watermarks: Fingerprints for diffusion images that are invisible and robust[J]. arXiv preprint arXiv:2305.20030, 2023.
>
> [7] Yang Z, Zeng K, Chen K, et al. Gaussian shading: Provable performance-lossless image watermarking for diffusion models[C]//Proceedings of the IEEE/CVF Conference on Computer Vision and Pattern Recognition. 2024: 12162-12171.

---

> > ### Comment · Reviewer_RwW9 · 2025-08-05
> > **My concerns are well addressed**
> >
> > Thanks for the detailed response! My concerns are well addressed.

---

> > > ### Author Response · Authors · 2025-08-06
> > >
> > > Dear Reviewer RwW9,
> > >
> > > Thank you very much for your time and insightful feedback on our submission. We truly appreciate your comments.
> > >
> > > We highly value any opportunities to address any potential remaining concerns before the discussion closes, which might be helpful for improving the rating of this submission. Please do not hesitate to comment upon any further concerns. Your feedback is extremely valuable!
> > >
> > >
> > > Sincerely,
> > >
> > > The Authors of Submission 15851

---

### Official Review · Reviewer_Ggtd · 2025-07-01

**Clarity:** 3
**Significance:** 3
**Originality:** 3
**Rating:** 4
**Confidence:** 3

**Summary:**

This paper proposes an INR-based watermarking framework designed for ultra-high-resolution images. The watermark can be accurately extracted from a single 128x128 image patch. To achieve this, the authors introduce a hierarchical multi-scale coordinate embedding mechanism and a low-rank watermark injection scheme. Experimental results demonstrate the effectiveness of the proposed method.

**Questions:**

Please refer to the weakness part.

**Ethical Concerns:**

["NO or VERY MINOR ethics concerns only"]

**Final Justification:**

A high-performance, efficient template-based watermarking scheme is valuable for both academic research and industrial implementation. Considering that, I tend to recommend accepting this paper.

**Limitations:**

The authors note limitations in visual quality and computational complexity.

**Paper Formatting Concerns:**

I have no other concerns.

**Quality:**

3

**Strengths And Weaknesses:**

### Strengths

1. The proposed INR-based template watermarking framework is novel and achieves strong performance.

2. The ability to reliably extract watermarks from only a small patch of high-resolution images is impressive.

3. The paper is well-written and presents a clear methodology.

### Weakness

1. From the visualizations, the watermark appears to vary across spatial locations. Does extraction accuracy depend on the location of the patch? If so, this could affect robustness.

2. The method section suggests that INR is only used to generate template watermarks and does not require per-image fitting. However, Section A.6 mentions that "the fitting process of INR can be computationally expensive," which limits embedding efficiency. This seems contradictory:  what exactly is being fitted during embedding, and why is it computationally intensive?

3. In Section 4.5, the authors mention that watermarks can be pre-generated, but in practice, precomputing $2^30$ unique watermarks is infeasible. It would be helpful to report the actual time required to generate a single watermark during embedding.

---

> ### Author Rebuttal · Authors · 2025-07-30
>
> Thank you for your patience in reviewing our work!
>
> We have noticed that you seem to have some questions about the computational efficiency of our method, and we sincerely apologise for any confusion caused by unclear wording.
>
> In the main text, we primarily aimed to highlight that INR has `weaker representational capabilities` compared to CNN. As a result, under the same scale, INR-based methods require longer training times and converge more slowly. However, since INR is essentially an MLP, its computational overhead during inference is significantly lower than that of CNN under the same scale. We will provide a detailed explanation in the following sections.
>
> # Q1: Robustness analysis across different spatial locations
>
> Due to our coordinate sampling strategy, our method can guarantee `decoding at any position`.
> To prove this, we embedded the watermark in 2K images (DIV2K dataset) and cropped blocks ranging from 128×128 to 1024×1024 pixels from five different positions (top left, bottom left, top right, bottom right, and center), then calculated the average accuracy rate of blocks(PSNR at 35 dB).
>
> |   Position   | 128x128 | 512x512 | 1024x1024 |
> | :----------: | :-----: | :-----: | :-------: |
> |   top left   |  99.93  |  99.99  |   99.99   |
> |  top right  |  99.83  |  99.99  |   99.99   |
> | bottom left |  99.87  |  99.99  |   99.99   |
> | bottom right |  99.83  |  99.99  |   99.99   |
> |    center    |  99.73  |  99.99  |   99.99   |
>
> As can be seen, our method can robustly decode regardless of the position.
>
> # Q2: Clarification on INR fitting process and computational cost
>
> We sincerely apologise for any confusion caused by our unclear statement.
>
> When we say that "the fitting process of INR can be computationally expensive," we mean that the `model converges` slowly during training. This is because INR has weaker representation capabilities than CNN, and our method requires more time to train than CNN-based methods. However, once training is complete, our method can not only pre-generate watermark templates, but also process arbitrary pixels independently and in parallel, resulting in high computational efficiency.
>
> Here is a comparison of training times using one RTX 3090 24G (32 batch):
>
> | Method | Proposed | HiDDeN | StegaStamp | MBRS | DWSF | TrustMark | $\mathrm{RAIM_{ARK}}$ |
> | :----: | :------: | :----: | :--------: | :--: | :--: | :-------: | :---------------------: |
> | Times |   26h   |  23h  |    19h    | 14h | 39h |     -     |    >20min per image    |
>
> Although our method is slower than methods such as MBRS, it is still faster than DWSF. Furthermore, $\mathrm{RAIM_{ARK}}$ requires fine-tuning for each image, which is unacceptable in large-scale production.
>
> # Q3: Report the actual time required to generate a single watermark during embedding
>
> Unlike CNNs, which require computations across the entire image, our method enables independent parallel processing for each pixel. Consequently, in theory, given sufficient GPU resources for parallel execution, our computational overhead is equivalent to the inference time of a single pixel.
>
> We demonstrate this by measuring the inference time for watermarking a single 2K image using one GPU, two GPUs(RTX3090 24GB):
>
> |      |    Ours(1 GPU)    |    Ours(2 GPU)    | DWSF | TrustMark | $\mathrm{RAIM_{ARK}}$ |
> | :---: | :---------------: | :---------------: | :--: | :-------: | :---------------------: |
> | Times | **24.16ms** | **13.74ms** | 74ms |   308ms   |    >20min per image    |
>
> Our INR method requires only 24.16ms on a single GPU, which is 3x faster than DWSF and 12.7x faster than TrustMark. Essentially, INR is just MLP, and its computational load is much lower than that of CNN of the same scale.

---

> > ### Comment · Reviewer_Ggtd · 2025-08-01
> >
> > Thanks for the rebuttal. My concerns are well addressed. I keep my original positive score.

---

> > > ### Author Response · Authors · 2025-08-06
> > >
> > > Dear Reviewer Ggtd,
> > >
> > > Thank you for your quick follow-up and for confirming that our rebuttal has addressed your concerns.
> > >
> > > We are very pleased to hear this and sincerely appreciate your positive assessment and support for our work. Your feedback throughout this process has been invaluable in helping us strengthen our submission.
> > >
> > > Thank you once again.
> > >
> > > Sincerely, The Authors of Submission 15851

---

### Official Review · Reviewer_2eV2 · 2025-07-01

**Clarity:** 3
**Significance:** 3
**Originality:** 2
**Rating:** 4
**Confidence:** 4

**Summary:**

This paper introduces a novel watermarking framework based on Implicit Neural Representations (INRs), explicitly designed for ultra-high-resolution (UHR) images. Traditional CNN-based watermarking methods often fail on UHR due to their high computational overhead and vulnerability to distortions such as scaling and cropping. In contrast, the proposed method decouples the watermark from the image resolution using a coordinate-based INR, enabling efficient, pixel-wise watermark embedding that is resolution-independent. Key contributions include a hierarchical multi-scale coordinate embedding, a low-rank watermark injection mechanism using CP decomposition, and a robust decoding strategy. Extensive experiments demonstrate superior robustness and efficiency compared to existing methods on datasets such as DIV2K, COCO, and FFHQ, even under extreme cropping and scaling scenarios.

**Questions:**

Could the INR watermarking be combined with content-aware features (e.g., via a hybrid INR-CNN model) to improve perceptual quality?

Is the embedding strength γ fixed, or could it be adaptively set based on local image content to balance robustness and perceptual quality?

How would the system perform against adversarial examples or model inversion attacks aimed at decoding or removing watermarks?

Can the authors demonstrate the robustness of the method on images shared via online platforms (e.g., Instagram, Twitter), where compression artifacts are non-differentiable?

Are there conditions under which the decoding accuracy drops significantly (e.g., heavy color jitter or combined augmentations)?

**Ethical Concerns:**

["NO or VERY MINOR ethics concerns only"]

**Final Justification:**

following the other reviews and rebuttal, I am happy to uplift my score to tend towards accept

**Limitations:**

The authors acknowledge lower visual quality due to template-based watermarking and the computational cost of INR fitting. Section 4.2 and Supplement A.6 discuss this in detail.

**Quality:**

2

**Strengths And Weaknesses:**

Strengths
Quality: The proposed method is well-engineered, with a sound methodology and strong experimental support. It demonstrates state-of-the-art robustness to distortion, particularly on high-resolution images.

Clarity: The paper is well-structured and includes detailed architectural diagrams (e.g., Figure 2) and a short ablation studies to support claims. However the fig 2 isn't the clearist figure to follow

Significance: The ability to embed and decode watermarks in UHR images under extreme distortions has practical significance in copyright protection and forensic tracking.

Originality: The use of INRs for watermarking, combined with coordinate-based sampling and low-rank injection, is novel. The low-rank decomposition for computational efficiency is an elegant solution.

Weaknesses
Visual quality trade-off: The method sacrifices some PSNR and SSIM compared to content-aware methods, which may be limiting in sensitive applications. This is the major concern I have


Real-world validation: While synthetic distortions are thoroughly tested, no results are presented for real-world scenarios like camera-captured images or social media uploads.

Limited discussion on potential adversarial attacks: There's minimal analysis on robustness against intentional tampering or adversarial watermark removal.

---

> ### Author Rebuttal · Authors · 2025-07-30
>
> Thank you for your patience and thorough review! We note your concerns regarding visual quality.
>
> It is important to emphasise that our `contribution` primarily lies in proposing a novel paradigm centred on INRs to address the challenge of robust embedding on ultra-large images.
>
> We adopt this architecture specifically to overcome the computational bottlenecks and memory constraints that CNN-based methods face with high-resolution inputs.
>
> The slight reduction in visual fidelity stems from the inherent nature of INRs, which process pixel coordinates independently and thus do not leverage local spatial features as effectively as CNNs.
>
> In fact, we made `considerable efforts` in perceptual quality but encountered some challenges. We will detail these in the following sections.
>
> # Q1: Using a hybrid INR-CNN model to improve perceptual quality
>
> We appreciate the reviewer's insightful suggestion. Indeed, we explored similar hybrid architectures during our initial investigations but found them unsuitable due to several fundamental challenges.
>
> we have explored hybrid architectures, using methods similar to LIIF [1], which involve first extracting features from the entire image using a lightweight CNN(3-layer Conv-BN-ReLU structure with 3×3 kernels), and then sampling these feature maps by coordinate for prediction with an INR.
>
> It can be expressed as:
>
> $$
> L_c = \mathrm{CNN}(I)
> $$
>
> $$
> z_e=\Gamma(L_c(x,y)\odot z_p)
> $$
>
> $L_c$ is similar to the embedded features mentioned in the section **"Hierarchical Multi-Scale Coordinates Embedding"** in the paper, while $z_e$ is equivalent to formula (4) in the paper.
>
> However, this introduces three `issues`:
>
> > (1) It is still necessary to perform CNN on the entire image, which results in computational overhead increasing with image size.
>
> > (2) Computational efficiency is extremely low. In the CNN part, we need to perform calculations on the entire image, but in the INR part, we only sample local regions for watermark embedding during training, resulting in most of the computational overhead being meaningless.
>
> > (3) More importantly, since the final watermark generation still relies on the INR, during actual training, the network `selectively ignores` features from the CNN, degenerating into pure INR training.
>
> If we only perform CNN on the specific regions where watermarks are to be embedded, This reintroduces the block-based generation problem shown in Figure 1 at our paper, requiring precise watermark block localization.
>
> we give our results on DIV2K dataset (2K resolution):
>
> |  Methods  | PSNR |  SSIM  | Avg ACC | GPU Memory |
> | :-------: | :---: | :----: | :-----: | :--------: |
> | Proposed | 37.27 | 0.9611 |  95.17  |   0.51GB   |
> | LIIF-Like | 38.14 | 0.9652 |  94.64  |  12.34GB  |
>
> Avg ACC represents the average decoding accuracy after various noise attacks, and GPU Memory represents the computing memory occupied by one sample during training. It can be seen that the visual quality has not improved significantly, while the GPU usage during training is much higher than that of our method.
>
>
> # Q2: About adaptive $\gamma$
>
> We have tried two strategies:
>
> > (1) Assigning a dynamic trainable parameter to each pixel, i.e., INR predicts not only RGB but also $\gamma$. However, this approach has issues where the predicted gamma values become completely ineffective, and due to the `lack of hard constraints`, this method produces very low visual quality.
>
> > (2) Pre-computing the texture richness of each pixel using the Laplacian operator and normalising it to [0,1] as $\gamma$. This approach effectively controls embedding only in high-frequency regions, but the decoding accuracy is very low. This is because INR cannot perceive the Laplacian operator, inevitably leading to `information loss` in low-frequency regions.
>
> This is results:
>
> | Methods | PSNR | SSIM | Avg ACC |
> |:-------:|:----:|:----:|:------:|
> | Proposed | 37.27 | 0.9611 | 95.17 |
> | Predict$\gamma$ | 22.53 | 0.7298 | 97.33 |
> | Laplacian Operator | 40.16 | 0.9811 | 78.41 |
>
> # Q3: Adversarial examples or model inversion attacks
>
> Currently, existing image watermarking methods do not extensively discuss this type of attack, so we primarily adopt mainstream attacks from previous SOTA methods like DWSF[2] (ACM MM 23) and $\mathrm{RAIM_{ARK}}$[3] (ACM MM 24) in our design.
>
> To our knowledge, **"model inversion attacks"** are mainly used to infer the original data used to train the model. This does not seem to be particularly relevant to the domain we are discussing.
>
> In addition, we test the performance of different **adversarial attack** methods using the DIV2K dataset, where watermarks are embedded at 2K resolution with a fixed PSNR of 35 dB.
>
>
> | Attack Method |   Proposed   |      |     DWSF     |      |   TrustMark   |      | $\mathrm{RAIM_{ARK}}$ |      |
> | :-----------: | :-----------: | :---: | :-----------: | :---: | :-----------: | :---: | :---------------------: | :---: |
> |              | Attacked PSNR |  ACC  | Attacked PSNR |  ACC  | Attacked PSNR |  ACC  |      Attacked PSNR      |  ACC  |
> |     PGD1     |     34.90     | 99.99 |     32.88     | 25.17 |     38.20     | 95.41 |          30.61          | 76.94 |
> |     PGD2     |     28.95     | 96.33 |     27.46     | 20.17 |     32.22     | 94.67 |          26.43          | 75.53 |
> |     FGSM1     |42.31     | 99.99 |     24.13     | 40.17 |     38.92     | 85.83 |          24.23          | 62.33 |
> |     FGSM2     |25.94 | 99.99 |7.6 | 41.83 |22.40 | 72.83 |7.74| 48.12 |
> |     BIM1     |36.92| 99.99 |18.75| 13.50 |33.04| 88.17 |25.12| 65.73 |
> |     BIM2     |31.44| 99.99 |13.39 | 13.17 |27.67| 84.32 |16.16 | 51.99 |
>
>
> We use **torchattacks** (a third-party adversarial example library) to test three different attack methods: PGD[4], FGSM[5], and BIM[6]. Attacked PSNR represents the PSNR of the distorted image and the clean image after the attack,  ACC is the accuracy rate after the attack.
>
> The specific settings are as follows:
>
> ```python
>     pgd1 = torchattacks.PGD(wrapped_model, eps=8/255, alpha=2/255, steps=20)
>     pgd2 = torchattacks.PGD(wrapped_model, eps=16/255, alpha=4/255, steps=20)
>     fgsm1 = torchattacks.FGSM(wrapped_model, eps=16/255)
>     fgsm2 = torchattacks.FGSM(wrapped_model, eps=128/255)
>     bim1 = torchattacks.BIM(wrapped_model, eps=64/255, alpha=8/255, steps=20)
>     bim2 = torchattacks.BIM(wrapped_model, eps=128/255, alpha=16/255, steps=20)
> ```
> wrapped_model is the watermark decoder from different methods.
>
> We find that our method demonstrates greater resistance to such attacks compared to other methods, as evidenced by two key factors:
> > (1) Under identical attack parameters, our method maintains a higher PSNR after the attack, indicating its insensitivity to adversarial perturbations;
>
> > (2) At the same PSNR level, our method achieves a higher accuracy rate, demonstrating its superior robustness against such attacks.
>
> # Q4: Robustness of online platforms
>
> We select 50 images and test our method on different media at PSNR 35 （2K resolution on DIV2K）:
>
> |          |Propsed| DWSF | $\mathrm{RAIM_{ARK}}$ | TrustMark |
> | :-------: | :-------------: | :---: | :---------------------: | :-------: |
> |  Twitter  | **92.45** | 60.67 |58.67|   81.27   |
> | Instagram | **99.67** | 76.33 | 66.13 |82.4|
> |  Wechat  | **99.33** | 70.53 |76.33|81.87|
>
> Our method is the most effective.
>
> # Q5: Performance under extreme conditions
>
> "We test the extreme cases with combined distortions involving `"noise + scaling + cropping"`.
>
> We first embed the messages into 2K images. These images are then subjected to various distortions. After that, we enlarge the distorted images to 4K resolution. Finally, we randomly crop 128×128 blocks from the enlarged images for decoding.
>
> |                        |    Identity    |       GN       |       GF       |      JPEG      |     Dropout     |    Rotation    |   Translation   |      Color      |
> | :---------------------: | :-------------: | :-------------: | :-------------: | :-------------: | :-------------: | :-------------: | :-------------: | :-------------: |
> |         propsed         | **85.95** | **87.97** | **58.07** | **52.10** | **87.40** | **77.87** | **74.97** | **76.70** |
> |          DWSF          |      50.02      |      49.9      |      50.73      |      50.16      |      49.16      |      50.73      |      50.12      |      50.16      |
> |        TrustMark        |      53.99      |      51.34      |      51.25      |      50.75      |      50.25      |      50.75      |      50.54      |      48.99      |
> | $\mathrm{RAIM_{ARK}}$ |   54.14 |    54.25      |      53.24      |      50.25      |      54.34      |      54.05      |      53.12      |      52.61      |
>
> In this case, our performance will decline significantly, but it will still be far better than other solutions.
>
> # Reference
>
> [1] Chen Y, Liu S, Wang X. Learning continuous image representation with local implicit image function[C]//Proceedings of the IEEE/CVF conference on computer vision and pattern recognition. 2021: 8628-8638.
>
> [2] H. Guo, Q. Zhang, J. Luo, F. Guo, W. Zhang, X. Su, and M. Li, “Practical deep dispersed watermarking with synchronization and fusion,” in Proceedings of the 31st ACM International Conference on Multimedia, 2023, pp. 7922–7932.
>
> [3] Y. Wang, X. Zhu, G. Ye, S. Zhang, and X. Wei, “Achieving resolution-agnostic dnn-based image watermarking: A novel perspective of implicit neural representation,” in Proceedings of the 32nd ACM International Conference on Multimedia, 2024, pp. 10 354–10 362.
>
> [4] Madry A, Makelov A, Schmidt L, et al. Towards deep learning models resistant to adversarial attacks[J]. arXiv preprint arXiv:1706.06083, 2017.
>
> [5] Goodfellow I J, Shlens J, Szegedy C. Explaining and harnessing adversarial examples[J]. arXiv preprint arXiv:1412.6572, 2014.
>
> [6] Kurakin A, Goodfellow I, Bengio S. Adversarial machine learning at scale[J]. arXiv preprint arXiv:1611.01236, 2016.

---

> > ### Author Response · Authors · 2025-08-06
> >
> > Dear Reviewer 2eV2,
> >
> > We sincerely appreciate your careful review and valuable time. As the discussion period draws to a close, we want to briefly follow up on our rebuttal.
> >
> > Our core contribution is a novel paradigm using Implicit Neural Representations (INRs) for robust watermarking on ultra-large images, rather than improving visual quality on top of existing methods. We acknowledge the slight trade-off in visual quality, which is an intentional design choice inherent to INRs. Our experiments provided in the rebuttal demonstrate this clearly.
> >
> > Moreover, the proposed method shows superior resilience against a wide range of challenges, from standard adversarial attacks to compression on real-world platforms, significantly outperforming current approaches.
> >
> > If our responses have adequately addressed your concerns, we hope that our clarification on this core contribution and the strong evidence of our method's robustness will justify improving our evaluation score.
> >
> > Thank you again for your time and consideration.
> >
> >
> > Sincerely,
> >
> > The Authors of Submission 15851

---

> ### Comment · Area_Chair_eKTJ · 2025-08-07
> **Please Read Author Response and Share Your Post-Rebuttal Comments**
>
> Dear Reviewer 2eV2,
>
> Thank you for your thoughtful and constructive reviews of this submission. Your time and expertise are greatly appreciated.
>
> The author response is now available. Please carefully read the rebuttal and update your reviews with post-rebuttal comments, indicating whether the response adequately addresses your concerns and whether it changes your assessment of the paper.
>
> Please also feel free to engage in discussion with your fellow reviewers, especially in cases where there are divergent opinions, so we can reach a more accurate and well-informed consensus. If you have any points of disagreement or clarification, this is a good time to raise and explore them collaboratively.
>
> Best regards,
>
> AC

---

> ### Author Response · Authors · 2025-08-09
>
> Dear Reviewer 2eV2,
>
> We sincerely appreciate your initial review of our submission and the time you have dedicated so far. As the discussion phase is entering its final hours, we kindly follow up to request your post-rebuttal feedback.
>
> Given the strong results and clarifications provided, we hope our rebuttal has sufficiently addressed your reservations. If so, we would be truly grateful if you could consider raising your assessment to reflect this, as your post-rebuttal input is essential to ensuring an informed consensus.
>
> All other reviewers have engaged in the discussion and explicitly confirmed that their concerns have been addressed. We would greatly value your updated perspective as well, so that the final decision can be made based on the most complete set of reviews.
>
> If you have any remaining concerns or questions about our rebuttal, we are more than willing to clarify them promptly before the discussion closes. Your timely input at this stage would be extremely helpful in ensuring that the consensus truly reflects all reviewers’ views.
>
> Thank you very much for your attention and consideration!
>
> Sincerely,
>
> The Authors of Submission 15851

---

### Author Response · Authors · 2025-08-09
**Thanks for All Reviewers**

Dear Reviewers,

We would like to extend our sincere gratitude to all of you for your time, effort, and insightful feedback on our submission. Your constructive comments have been invaluable in helping us strengthen our work. We are also deeply encouraged by your recognition of our paper's `novelty` and `potential significance`.


# Reiteration of Core Contributions
In response to your valuable questions, we wish to briefly reiterate the core contributions of our work.

Our work introduces a `pioneering application of INRs` to the field of `ultra-high resolution image watermarking`, specifically designed to overcome the fundamental bottlenecks of existing CNN-based methods.

Our framework achieves unprecedented scalability through **resolution-independent coordinate sampling**, making memory usage constant regardless of image size. We ensure robustness by enriching these coordinates with a **hierarchical multi-scale embedding** to learn complex, distortion-resistant patterns. For model efficiency, a **low-rank watermark injection** strategy dramatically reduces parameter count.

We have summarised our `strengths`:

> `Unmatched Efficiency`: We are the novel watermarking method that can process UHR images with constant memory usage. Traditional CNN methods crash due to GPU memory overflow on large images, while our approach fundamentally solves the UHR computing bottleneck.

> `Extreme Robustness`: Our core competitiveness lies in our ability to resist extreme cropping and scaling attacks. Even if only less than 1% of the image remains, we can still successfully decode the watermark information with an accuracy rate of over 98%. In the most rigorous tests, such as social media dissemination and mixed noise attacks, our method remains robust, proving its reliability in complex real-world environments.

> `Large-Scale Applications`: Our watermarks can be pre-generated offline and reused on a massive scale, making our method ideal for large image libraries, news agencies, content distribution platforms, and other scenarios that require the rapid addition of copyright information to millions of images.


# Summary of Discussion & Sincere Plea

In our rebuttal, we have done our utmost to address every concern raised, providing detailed explanations and a significant number of new experiments. We sincerely hope that our responses and the new data provided have fully resolved your initial reservations.


Moreover, we are sincerely encouraged that for many of you, our responses and the new data successfully `resolved your initial reservations`, as you kindly confirmed during the discussion period.


Given the efforts we have made to clarify these points and the strong results from our supplementary experiments, we kindly ask that you fully consider our rebuttal and the subsequent discussion in your final decision. We are hopeful that our work and responses will merit a `positive re-evaluation` of our submission.

Thank you once again for your valuable time and professional contributions to this review process.

Sincerely,

The Authors of Submission 15851

---

### Decision · Program_Chairs · 2025-09-17

**Decision:**

Accept (poster)

**Comment:**

This paper presents an INR-based ultra-high-resolution watermarking framework, a novel approach combining resolution-independent coordinate sampling, multi-scale embedding, and low-rank watermark injection to enable robust watermark extraction from small patches and resist extreme cropping and scaling in high-resolution images. In the initial concerns, reviewers agreed the work is novel, practical, and relevant, especially its use of INRs for robustness under UHR conditions. Main concerns include slightly lower visual quality, limited evaluation (e.g., mixed distortions, real-world channels, adversarial attacks), and clarity on capacity and efficiency.

In rebuttal, the authors added new experiments and clarified efficiency/capacity trade-offs. These responses addressed the key issues, leading most reviewers to maintain or increase their scores, with consensus shifting to acceptance. The AC has also read the comments by reviewer and the corresponding rebuttal and discussions, and agree that the remaining concerns do not suffice for rejection. Thus, I recommend acceptance of this paper.